# UK Hydrological Outlook using Historic Weather Analogues

Wilson Chan[1*], Katie A. Facer-Childs[1], Maliko Tanguy[1,2], Eugene Magee[1], Burak Bulut[1], Nicky Stringer[3], Jeff Knight[3], Jamie Hannaford[1,4]

[1]UK Centre for Ecology & Hydrology (UKCEH), Wallingford, OX10 8BB, England
[2]European Centre for Medium-Range Weather Forecasts, Reading, RG2 9AX, England
[3]Met Office, Exeter, EX1 3PB, England
[4]Irish Climate Analysis and Research UnitS (ICARUS), Maynooth University, Maynooth, Co. Kildare, Ireland

*Correspondence to*: Wilson Chan (wilcha@ceh.ac.uk)

**Abstract**

Skilful seasonal hydrological forecasts are beneficial for water resources planning and disaster risk reduction. The UK Hydrological Outlook (UKHO) provides river flow and groundwater level forecasts at the national scale. Alongside the standard Ensemble Streamflow Prediction (ESP) method, a new Historic Weather Analogues (HWA) method has recently been implemented. The HWA method samples within high resolution historical observations for analogue months that matches the atmospheric circulation patterns forecasted by a dynamical weather forecasting model. In this study, we conduct a hindcast experiment using the GR6J hydrological model to assess where and when the HWA method is skilful across a set of 314 UK catchments for different seasons. We benchmark the skill against the standard ESP and climatology forecasts to understand to what extent the HWA method represents an improvement to existing forecasting methods. Results show the HWA method improves river flow forecasts most notably in winter, with skilful winter river flow forecasts now possible across the UK compared to the standard ESP method where skilful forecasts were only possible in southeast England. Winter river flow forecasts using the HWA method were also more skilful in discriminating high and low flows across all regions. Catchments with the greatest improvement tended to be upland, fast responding catchments with limited catchment storage and where river flow variability is strongly tied with climate variability. Skilful winter river flow predictability was possible due to relatively high forecast skill of winter atmospheric circulation patterns and the ability of the HWA method to derive high resolution meteorological inputs suitable for catchment hydrological modelling. However, skill was not uniform across different seasons. Improvement in river flow forecast skill for other seasons was modest, such as moderate improvements in northern England and northeast Scotland during spring and little change in autumn. Skilful summer flow predictability remains possible only for southeast England and skill scores at some catchments were reduced compared to the ESP method. This study demonstrates that the HWA method can leverage both climate information from dynamical weather forecasting models and the influence of initial hydrological conditions. An incorporation of climate information improved winter river flow predictability nationally, with the advantage of exploring historically unseen weather sequences. The strong influence of initial hydrological conditions contributed to retaining year-round forecast skill of river flows in southeast England. Overall, this study provides justification

for when and where the HWA method is more skilful than existing forecasting approaches and confirms the standard ESP method as a "tough to beat" forecasting system that future improvements should be tested against.

## 1 Introduction

Seasonal streamflow forecasting is a valuable tool in water resources and disaster risk management. In the UK, seasonal streamflow forecasts are used by the Environment Agency, private water companies, the agricultural industry, and others in the environment and leisure sectors, to assess the risk of oncoming and ongoing drought, as well as the potential risk during

flood-prone seasons. Water resource decisions such as reservoir operations and water transfer schemes can benefit from seasonal streamflow forecasting products, as well as prospects for irrigation and groundwater abstractions. The UK Hydrological Outlook (UKHO) has been operational since 2013, providing monthly and seasonal forecasts of river flows and groundwater levels (Prudhomme et al., 2017), and undergoes continual assessment and development. The skill of seasonal hydrological forecasting systems is typically lower than short-range forecasts that are used for flood event prediction. Seasonal

river flow predictability derives mainly from the initial hydrological conditions (e.g. soil moisture and groundwater storage and current river levels) and the predictability of weather and climate over the forecast period (Wood et al., 2019). The relative contributions of these two factors for river flow predictability varies between different catchments for different lead times (Li et al., 2009; Wood and Lettenmaier, 2008).

### 1.1 Drivers of UK rainfall

The UK is located in the mid-latitude belt of predominantly westerly winds. There are strong windward and leeward effects in both rainfall and streamflow (Svensson and Jones, 2002). Rainfall is higher in the north and west because of the combined effect of orography and exposure to the humid westerly winds. Here, autumn and winter makes the largest contribution to the annual total, because the most frequent and intense depressions occur during these seasons. In contrast, the sheltered south and east of the country receives less rainfall, and the distribution through the year is more even (Hulme and Barrow, 1997).

In winter, the southeast to northwest rainfall and streamflow gradients are typically enhanced during a season with a positive North Atlantic Oscillation (NAO) index (Svensson et al., 2015). The NAO is the leading mode of variability in the North Atlantic, characterised by a seesaw in pressure between the subpolar Icelandic low and the subtropical Azores high regions of the North Atlantic Ocean. For a positive winter NAO, there is a clear northward shift and intensification of the North Atlantic storm track. The NAO accounts for at least half of the seasonal rainfall variability in the northern British Isles and Scandinavia

(Seager et al., 2020). A positive NAO establishes low-level westerly or south-westerly time-mean flow from the eastern North America to Scandinavia, resulting in increased rainfall in northwest Europe as the enhanced southwesterlies and associated storms meet topography. For the UK, a positive NAO is generally associated with wetter than average conditions, but there

are substantial regional differences. Rainfall over western UK is strongly positively correlated with the winter NAO but this relationship is weak or marginally reversed for southeastern UK with these regions seeing a slight decrease in rainfall during a positive NAO phase. In contrast, a negative winter NAO is associated with negative rainfall anomalies for the UK as a whole. Regionally, a negative winter NAO is negatively correlated with rainfall across western UK but marginally associated with wetter than average conditions in parts of eastern and central England (West et al., 2019).

In summer, the NAO influences UK rainfall and is associated with the North Atlantic jet latitude. Its pattern is north-shifted compared to its wintertime counterpart. A positive summer NAO (commonly referred to as SNAO) is characterised by the jet stream passing far to the north of the UK and thus drier than average conditions. A negative SNAO is associated with a more southerly jet position close to the UK and results in wetter than average conditions (Folland et al., 2009, Dunstone et al., 2018). However, the total summer rainfall variability explained by the SNAO is smaller than its winter counterpart, and could also be influenced by large internal climate variability, soil moisture anomalies, sea surface temperatures and other modes of climate variability (e.g. East Atlantic EA pattern) (e.g. West et al., 2021; Wilby et al., 2004).

From a hydrological perspective, catchments across the UK differ in river flow response to climate variability. Catchments in southeastern UK can be either slowly or quickly responding to rainfall. Groundwater dominated catchments on permeable geology (particularly chalk) can have a streamflow response delayed by months (e.g. Chiverton et al., 2015) since groundwater provides a large reservoir to feed rivers. Small catchments on impermeable clay, however, do not have similar large reservoirs and hence may respond much more quickly, even within a matter of hours. Catchments in the north and west are more homogeneous, with hilly and/or impermeable catchments being predominantly fast-responding. Significant positive relationships were found between the winter NAO and elevated river flows in northwest UK but a very weak response was found for catchments in the southeast, partly attributed to regional differences in rainfall as outlined above but also due to the effects of physical catchment characteristics which further affect rainfall-river flow propagation (West et al., 2022). In the summer, the influence of the SNAO on river flows is weaker than winter but are more spatially coherent with the phase of the EA pattern playing a moderating role which strengthen or reduce rainfall and hence river flows (West et al., 2021). Summer river flows are also highly influenced by antecedent conditions, including lagged responses to wintertime river flows and summer atmospheric circulation indices (Svensson and Prudhomme, 2005; Wilby et al., 2004).

## 1.2 Methods of Seasonal Streamflow Prediction

Existing approaches for river flow forecasting range from dynamical weather forecasting models to statistical approaches and can broadly be categorised into four main strands. First, deterministic statistical methods such as the flow persistence method generates river flow forecasts by repeating the flow anomaly in the most recent month. Similarly, the flow analogue method uses historical river flows sequences that are most similar to the recent past as the forecast (Quinn et al., 2021; Svensson,

2016). An assessment of the skill in the UK showed that skilful forecasts using the persistence/flow analogue forecasts can be made for slow-responding catchments in southeast England with large storage capacities (Svensson, 2016). Skilful persistence forecasts were similarly found for lowland, permeable catchments in the Republic of Ireland by Quinn et al. (2021).

Second, ensemble probabilistic river flow forecasts can be made with information from historical climate. The most common technique is the Ensemble Streamflow Prediction (ESP) approach, where ensemble forecasts are made by assuming the repetition of meteorological traces (rainfall, potential evapotranspiration and temperature) from historical years (Day, 1985). ESP is widely used for streamflow forecasting, including in the UK Hydrological Outlook. The source of skill for ESP derives from initial hydrological conditions (e.g. antecedent catchment storage in soils, aquifers and snowpack) which are often predictable at long lead times (Wood and Lettenmaier, 2008). ESP skill varies across the UK, with the highest skill at catchments underlain by permeable aquifers with high catchment storage capacities in the south and east, which show skilful forecasts up to a year ahead in some cases (Harrigan et al., 2018). Fast-responding catchments with low storage capacities in the north and west show lowest skill which drops rapidly with lead time. ESP skill also varies with seasons, with forecasts in the winter and autumn months having highest skill for high storage catchments in south/east. ESP represents a low-cost and efficient approach in the absence of skilful meteorological forecasts and are often used as a benchmark forecast for other forecasting approaches to compare against (Harrigan et al., 2018).

The third approach for streamflow forecasting is stylised scenarios such as creating plausible river flow trajectories using hydrological models by assuming rainfall over the next month or season matches a pre-selected percentage of the long-term average (LTA) (e.g. 60 %, 80 %, 100 % and 120% of LTA). The UK Environment Agency uses this method in their monthly Water Situation Reports (e.g. Environment Agency, 2022) as a low-cost method to provide a forward look at the magnitude of river flows that can be expected given specific rainfall volumes. The same approach is also used by UK water companies to provide a forward outlook and bears similarity with a more routine exploration of plausible worst cases. For example, large ensembles of initialised climate model simulations (e.g. pooled hindcasts from seasonal forecasting systems or decadal prediction systems) have been employed to provide physically plausible weather sequences to understand plausible extreme events that may be more severe than what has been historically observed (Chan et al., 2024; Kay et al., 2024; Thompson et al., 2017).

Finally, probabilistic ensemble river flow forecasts can also be made by directly using output from numerical weather prediction models (NWP) or climate models. For example, the Global Flood Awareness System (GloFAS) couples the ECMWF Integrated Forecasting System (IFS) ensemble meteorological forecasts with the LISFLOOD hydrological model to generate river flow forecasts at the global scale on a daily basis (Harrigan et al., 2023). A similar approach has been applied operationally within the UKHO by using a spatially distributed monthly water balance model driven by national-scale or regionally averaged monthly rainfall forecasts from the Met Office GloSea forecasting system (Bell et al., 2017). However,

statistical or computationally intensive dynamical downscaling methods would be required to reconcile coarse scale outputs from NWP models before they can be used for detailed catchment hydrological modelling. Hence, most past applications in the UK rely on statistically downscaling or spatially and temporally disaggregating forecast data.

Different types of hydrological models can be applied with any of the above river flow forecasting methods. Hydrological models can generally be categorised as conceptual, process-oriented/physically-based and data-driven. First, conceptual models, where hydrological processes are parameterised through a series of storage components, are widely use given its low data requirements and computational efficiency. Conceptual models are used in river flow forecasting in the UK (e.g. Harrigan et al., 2018). Second, process-oriented models aims to simulate hydrological processes based on fundamental physical laws but are data intensive and more computational demanding (e.g. US National Water Model Cosgrove et al., 2024). Third, data-driven approaches such as machine learning models to simulate river flows (e.g. LSTMs - Kratzert et al., 2019 and Lees et al., 2021 for Great Britain) are emerging and hybrid approaches combining traditional hydrological models with machine learning post-processing (e.g. Slater et al., 2023) are also increasingly used. Although not yet applied operationally in the UK, machine learning methods have been applied to improve hindcast skill over a range of time scales from monthly river flow forecasts (e.g. Akbarian et al., 2023) to decadal flood prediction (e.g. Moulds et al. 2023).

## 1.3 Conditioned ESP approaches

Several studies have demonstrated that forecast skill can be improved by incorporating climate information in ESP forecasts, often referred to as conditional ESP. As catchment precipitation and temperature (and river flows) are affected by large-scale modes of climate variability,, conditional ESP forecasts can be made by sub-sampling only the meteorological traces that share the current climate states (e.g. current El Nino-Southern Oscillation (ENSO) phase) (Mendoza et al., 2017). Following this method, Wood and Lettenmaier (2006) found beneficial forecast improvements for catchments in western United States, particularly during strong ENSO phases with similar improvements in forecast skill of 5-10% found by Beckers et al. (2016) in the Columbia River basin. An alternative to sub-sampling ESP forecasts is to assign weights to ESP traces in post-processing based on similarity with atmospheric circulation information of the recent past (Baker et al., 2021; Mendoza et al., 2017). In Europe, studies have shown that the forecast skill for the winter NAO has improved substantially and shows useful levels of predictability at seasonal lead times (Scaife et al., 2014; Smith et al., 2020). Recent research further demonstrated that the influences of global teleconnection patterns (e.g. those from ENSO) on the NAO can be predicted with long lead times (up to 1 year) (Scaife et al., 2024). This opens up the opportunity to leverage the improved predictability in atmospheric circulation patterns to improve the skill of river flow forecasts, especially in catchments where a strong positive relationship between climate variability and river flows are found. Stringer et al. (2020) demonstrated a Historic Weather Analogues (HWA) approach to generate winter rainfall and temperature forecasts by sub-selecting independent "analogue" months from the

historical observations that resembles spatially the seasonal dynamical signals (i.e. atmospheric circulation patterns) forecasted by the Met Office GloSea system. While conditioned ESP methods rely on sub-sampling or weighting historical traces based on large-scale climate signals, the HWA approach further advances this concept by identifying specific historical weather patterns that closely match forecasted atmospheric circulation states. This enables forecasts to more directly leverage reliable dynamical model outputs and can provide higher spatial resolution than traditional ESP-based methods. This approach thus takes advantage of forecasted atmospheric circulation characteristics which may be more reliable than the direct forecasts of rainfall and considers probabilistic weather regime forecasts (i.e. likelihood of specific atmospheric circulation configurations at different lead times) (e.g. Richardson et al., 2018). The HWA approach also enables forecasting at higher spatial resolutions as high-resolution observations (e.g. 1km) can be used to generate analogue forecasts.

The HWA approach is operationalised as part of the UKHO as a hybrid meteorological-based forecast method that complements the standard ESP approach and the data-driven persistence/analogue methods. This paper aims to conduct a catchment-based investigation of the forecast skill of the HWA method within a hindcast experiment over the 1993-2016 hindcast period using a catchment hydrological model. A companion paper – Rhodes-Smith et al. (in review), aims to analyse the skill of the HWA forecasts with an alternative hydrological modelling system, a nationally parameterised, spatially distributed hydrological model also used within the UKHO. In this paper, we aim to address the following specific research questions:

1. When is the HWA forecasts skilful, across different seasons?
2. Where are the HWA forecasts skilful, across UK catchments and regions?
3. To what extent are the HWA forecasts an improvement against the standard ESP approach?

## 2 Methods

### 2.1 Data

A total of 314 catchments from the National River Flow Archive (NRFA) were chosen for this study (Figure 1). The same set of catchments is used operationally within the UKHO. The catchments were chosen to represent a wide range of hydroclimatic conditions across the UK, spanning "flashy" fast-responding upland catchments in the north to slow-responding, lowland catchments in the southeast underlain by Chalk and Limestone aquifers. The catchments include those that are studied by the National Hydrological Monitoring Programme to assess river flow trends and past floods and droughts. Out of the 314 catchments, 128 catchments are part of the UK Benchmark Network (Harrigan et al. 2017) that are relatively free from human influences. The catchments spans nine hydroclimate regions based on climatological and hydrological similarity (National River Flow Archive, 2014) (Figure 1). Table 1 shows selected statistics across the 314 catchments and their hydroclimate regions. Several derived rainfall/flow statistics (Standardised Annual Average Rainfall - SAAR and Baseflow Index – BFI)

and physical catchment characteristics (catchment wetness index) were extracted from the NRFA database via the *rnrfa* R package (Vitolo et al., 2016). BFI is a measure of catchment storage with typical ranges between 0.15 (fast responding

impervious catchments underlain by clay) to 0.9 (slow-responding Chalk streams with a high baseflow component). SAAR refers to the mean annual rainfall over each catchment for the 1961-1990 period and the catchment wetness index is a measure describing the proportion of time soils in the catchment are defined as wet (i.e. low soil moisture deficits). Several atmospheric circulation indices were also computed. The NAO index is taken from University of East Anglia's Climatic Research Unit (CRU) and is based on the mean sea level pressure difference between the Azores and Iceland using a 1951-1980 reference

period. A more direct characterisation of atmospheric circulation is obtained by the North Atlantic jet stream intensity for the hindcast period, calculated using zonal wind speeds taken from the ERA5 reanalyses following Woollings et al. (2010) using the *jsmetrics* python package (Keel et al., 2024) .

Observed daily river flow (m³/s), total rainfall (mm/day), and potential evapotranspiration (PET) (mm/day) were extracted for

each catchment as input for catchment hydrological modelling. Observed rainfall and temperature was taken from the 1km HadUK-Grid dataset (Hollis et al., 2019) and PET was calculated using daily average temperature with the McGuinness-Bordne equation calibrated specifically for the UK (Tanguy et al., 2018). Previous comparison between the temperature-based McGuinness-Bordne equation and the "reference" Penman-Monteith equation showed satisfactory performance and was ranked the best out of seven alternative temperature-based equations (Tanguy et al., 2018). While historical gridded PET

calculated using the Penman-Monteith equation are available across the UK, they are not updated in real time and the operational UKHO relies on the McGuiness-Bordne equation to estimate PET.

**Table 1 Summary statistics of the 314 selected catchments for the UK across nine hydroclimate regions. The range given for each catchment characteristic is the 5th and 95th percentile across catchments in each region. Catchment characteristics are retrieved**
**from the UK National River Flow Archive (NRFA). RR is runoff ratio and Fs is fraction of precipitation falling as snow (across hydrological years 1983-2014). Reproduced with permission from Harrigan et al. (2018).**

| Region | n | Area (km2) | Median elevation (m a.s.l.) | BFI (−) | Mean annual Q (mm yr−1) | Mean annual P (mm yr−1) | Mean annual ETp (mm yr−1) | RR $\bar{Q}/\bar{P}$ (−) | $\bar{F}$s (−) |
|---|---|---|---|---|---|---|---|---|---|
| UK | 314 | 181 (27, 1844) | 179 (60, 437) | 0.5 (0.27, 0.89) | 595 (162, 1839) | 1031 (648, 2202) | 504 (400, 542) | 0.59 (0.24, 0.87) | 0.03 (0.01, 0.14) |
| WS | 35 | 229 (64, 1745) | 268 (146, 468) | 0.33 (0.20, 0.61) | 1115 (554, 2847) | 1460 (998, 3145) | 428 (391, 476) | 0.74 (0.58, 0.90) | 0.06 (0.03, 0.12) |
| ES | 43 | 289 (70, 2759) | 303 (100, 596) | 0.51 (0.34, 0.67) | 693 (338, 1498) | 1040 (783, 1970) | 432 (387, 481) | 0.63 (0.44, 0.84) | 0.09 (0.06, 0.21) |

| | | | | | | | | | |
|---|---|---|---|---|---|---|---|---|---|
| NEE | 30 | 344 (11, 1910) | 264 (88, 449) | 0.43 (0.26, 0.82) | 559 (344, 1054) | 1037 (757, 1462) | 486 (455, 516) | 0.57 (0.44, 0.83) | 0.07 (0.04, 0.09) |
| ST | 25 | 198 (48, 6345) | 145 (87, 312) | 0.56 (0.41, 0.79) | 392 (209, 844) | 858 (670, 1311) | 511 (493, 528) | 0.46 (0.31, 0.68) | 0.03 (0.02, 0.05) |
| ANG | 33 | 99 (23, 1540) | 80 (33, 132) | 0.56 (0.25, 0.88) | 183 (128, 254) | 655 (600, 716) | 535 (528, 551) | 0.27 (0.21, 0.36) | 0.03 (0.03, 0.04) |
| SE | 59 | 134 (18, 1091) | 105 (43, 178) | 0.64 (0.23, 0.96) | 356 (146, 568) | 856 (654, 1033) | 529 (520, 541) | 0.42 (0.20, 0.64) | 0.02 (0.01, 0.03) |
| SWESW | 47 | 174 (29, 915) | 207 (77, 377) | 0.51 (0.37, 0.67) | 979 (507, 1549) | 1372 (1002, 1971) | 519 (495, 537) | 0.69 (0.51, 0.83) | 0.01 (0.00, 0.03) |
| NWENW | 32 | 112 (30, 1094) | 210 (108, 360) | 0.35 (0.27, 0.58) | 1154 (390, 2102) | 1529 (884, 2429) | 478 (457, 514) | 0.75 (0.44, 0.91) | 0.04 (0.02, 0.05) |
| NI | 10 | 230 (68, 1235) | 140 (90, 172) | 0.38 (0.33, 0.50) | 688 (533, 1206) | 1111 (917, 1565) | 475 (466, 488) | 0.63 (0.57, 0.77) | 0.01 (0.00, 0.02) |

## 2.2 Hydrology Model

The GR6J (Génie Rural à 6 paramètres Journalier) hydrological model version 1.0.2 initialised using the *airGR* R package (Coron et al. 2017) was selected to simulate river flows across the observational period and generate the river flow hindcast archive. GR6J is a daily lumped catchment hydrological model consisting of six model parameters for model calibration (Pushpalatha et al., 2011). It is widely used for both streamflow forecasting and water resources planning in the UK, including within the UKHO and by many water companies (e.g. Anglian Water Drought Plan 2022). GR6J shows reliable simulation of

river flows across a diverse range of UK catchments and is particularly attractive given its parsimonious nature given the need for computational efficiency with a large number of model runs. GR6J was developed from the original four parameter variant GR4J model with the addition of two parameters designed to improve simulation of low flows, particularly at slow-responding catchments with a strong baseflow groundwater contribution (Pushpalatha et al., 2011). The operational UKHO recently switched from GR4J to GR6J to align with industry norms and a comparison of ESP forecast skill between the two models

show minimal differences for most catchments across the UK[1]. The input data required for GR6J is daily catchment-averaged rainfall and PET and the model was calibrated against observed river flows from the UK National River Flow Archive (NRFA)

---

[1] Details of implications to ESP forecast skill from the switch from GR4J to GR6J hydrological model within the operational UK Hydrological Outlook can be found in the following website: https://hydoutuk.net/about/methods/river-flows

with the in-built automatic calibration procedure (Coron et al., 2023). The automatic calibration procedure uses a steepest local search algorithm, and the modified Kling-Gupta efficiency calculated from square root transformed simulated river flows (KGE2) was chosen as the objective function (Kling et al., 2012). Observed river flows over the 1961-2023 period were used for hydrological model calibration and the top performing parameter set according to KGE2 was subsequently used to simulate river flows over the baseline period (retrospective simulation). The calibrated GR6J model results over the baseline period showed satisfactory model performance across the UK compared to observed river flows. The KGE2 score for the top performance parameter set across all selected catchments is shown in Figure 1 and model performance is generally comparable or better than other lumped catchment models applied for the UK (e.g. Lane et al., 2019).

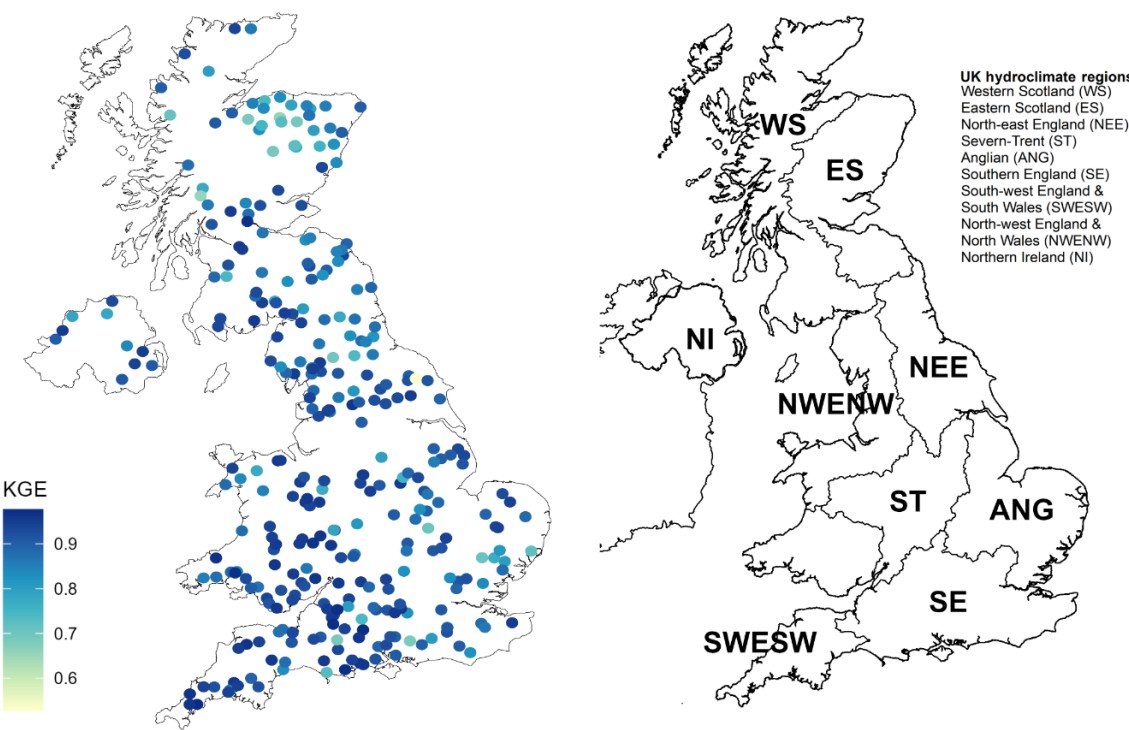

**Figure 1 Model performance based on the modified Kling-Gupta efficiency (KGE) calculated from square root transformed simulated river flows (left) and the nine UK hydroclimate regions used to aggregate and summarise skill score results (right).**

## 2.3 Generating Historic Weather Analogues

The historic weather analogues are generated based on the operational GloSea6 seasonal prediction system. GloSea6 provides 1- and 3-month forecasts every month for the operational UKHO. GloSea6 is based on a version of the Met Office Hadley

Centre climate model (HadGEM3) with an atmospheric resolution of 0.83° x 0.55° and 85 vertical layers, and a 0.25° ocean resolution with 75 vertical layers (MacLachlan et al., 2015). The operational system provides seasonal forecasts (up to 6-months) from 2 simulations initialised at 0 UTC each day. Ensemble forecasts are obtained by pooling members across 3 weeks (21 days) of initialisation times, resulting in an ensemble size of 42 members. In this study, retrospective forecasts ('hindcasts') for each meteorological season over the 1993-2016 period are produced. The hindcasts are initialised from a subset of dates (1st, 9th and 17th) each month as previously detailed in Stringer et al. (2020). In total, we included 17 hindcast ensemble members, pooled across the aforementioned three initialisation dates, giving a 51-member ensemble for each season. The number of hindcast ensemble members and initialisation dates were chosen to resemble the operational system as closely as possible, and the resulting 51-member ensemble is similar in size to operational forecasts, thus providing a fair reflection of operational forecast skill. An important purpose of the hindcast is to estimate the skill of the forecast system, as verification accrues too slowly to make such determinations from real-time forecasts alone. Here, we use these hindcasts, together with hydrological models, to estimate skill in forecasting the hydrological conditions described in the UKHO. Hindcasts include the four conventional meteorological seasons (DJF - winter, MAM - spring, JJA - summer and SON - autumn).

Traditionally, output from seasonal meteorological forecasting systems are averaged across the three-month season. In addition, the spatial resolution of the global climate model is relatively coarse. Hydrological models within the UKHO require daily and high spatial resolution meteorological input (e,g, 1km gridded rainfall and temperature). Stringer et al. (2020) developed a means of downscaling outputs from seasonal forecast systems to provide suitable spatiotemporal resolution inputs for use in UK hydrology modelling. The key to this methodology is matching hindcast member atmospheric circulation (as measured by the mean sea level pressure (MSLP) pattern) with the average circulation pattern for periods in historical records from 1960. This analogue approach is applied to the individual months of the 3-month season, so each analogue season can be comprised of sequences of days drawn from different years. By this means, the pool of possible seasonal analogues is substantially increased, and more extreme seasons than are contained in the observational record can be achieved (Figure 2). By finding real months that are analogues of predicted monthly patterns, high-resolution records of observed daily UK climate can be used to infer plausible scenarios of rainfall and temperature at the local scale and at daily resolution. For each of the 51 original GloSea6 members, we construct 10 analogue-based seasonal sequences of 1 km resolution daily meteorological variables for use in hydrological modelling. The "best" 10 analogues are selected based on the smallest Euclidian distance between the MSLP patterns of real months and the predicted monthly patterns over the North Atlantic-European domain centred on the UK, giving a total of 510 analogue ensemble 'members' for each forecast member (i.e. 10 analogues x 51 GloSea6 ensemble members).

Seasonal forecast systems often exhibit a 'signal-to-noise' problem in which the modes of climate variability are skilfully predicted but with an amplitude that is too small to be consistent with this level of skill (Eade et al., 2014; Scaife and Smith, 2018). A further aspect of the HWA approach as described in Stringer et al. (2020) is that for winter (DJF) the GloSea6

circulation patterns used in the generation of analogues are modified to increase the amplitude of the North Atlantic Oscillation (NAO) component of the circulation pattern. For winter (DJF), the NAO is inflated by a factor of approximately 2 prior to analogue selection. This overcomes the lack of signal amplitude in many NAO-related variables, including rainfall, which

would otherwise tend to make the predicted statistical distributions of rainfall too similar from year to year. Overall, Stringer et al. (2020) showed that the ensemble mean winter rainfall forecasts using the HWA approach is well correlated with observations over the 1993-2016 hindcast period, particularly for western UK and Northern Ireland which have greater influence from predictable NAO variability. Figure S1 shows correlation of HWA ensemble mean hindcast rainfall and HadUK-Grid observed rainfall for all seasons and each catchment. The ensemble mean hindcasts show good correlation with

observed rainfall across northern and western areas for winter and spring and moderate correlation for northern and southwest England in autumn. Hindcast rainfall is least well correlated with observed rainfall in the summer where large parts of the UK show negative correlations, suggesting low seasonal forecast skill of atmospheric circulation and drivers of rainfall in the summer (Weisheimer and Palmer, 2014). A national-scale comparison of the HWA rainfall forecasts and the HadUK-Grid observed rainfall at 1km, including at ungauged locations is included in the companion paper: Rhodes-Smith et al. (in review).

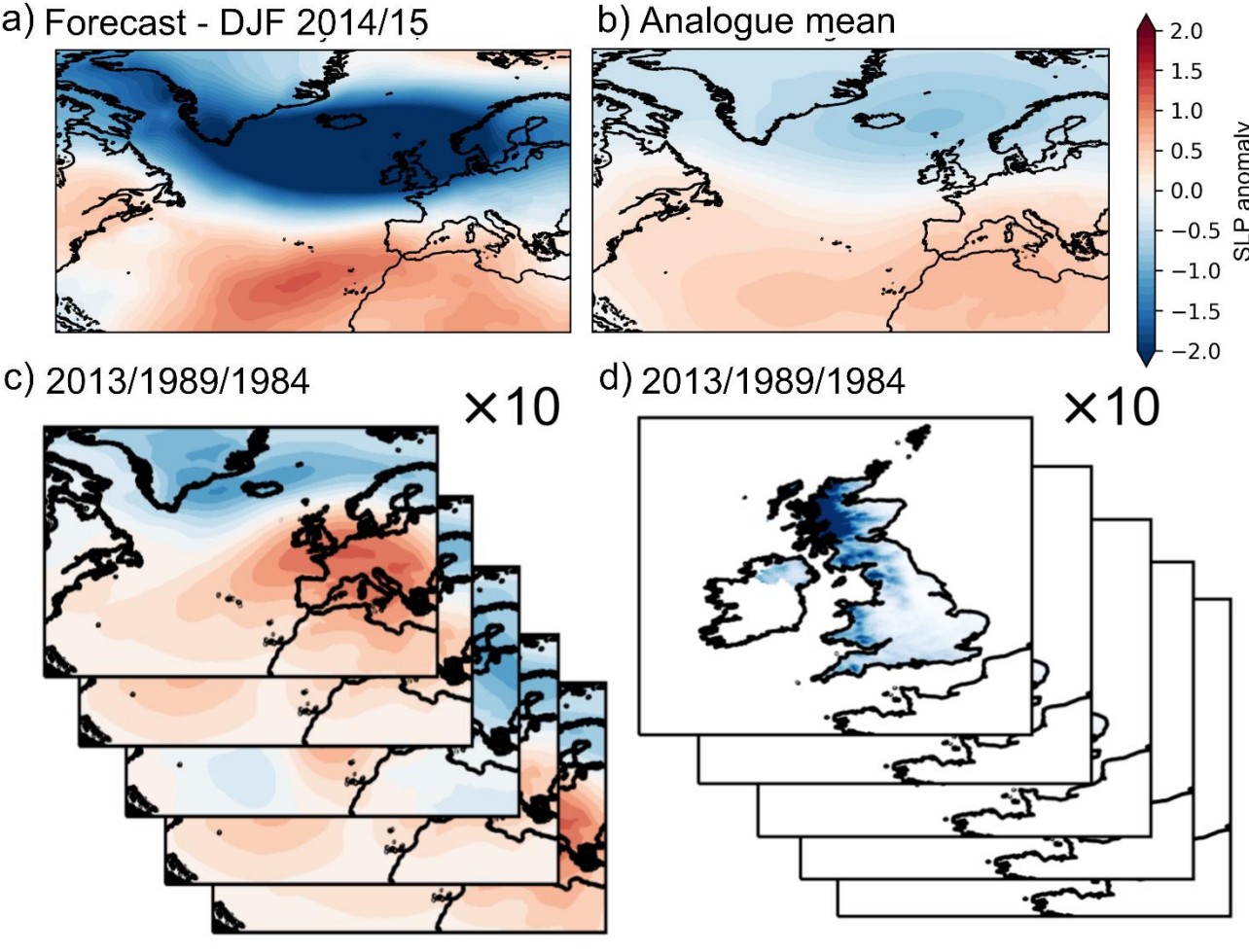

**Figure 2: Schematic outlining the Historic Weather Analogues method for an example winter (2014/15). Panel a) shows GloSea forecast of standardised sea level pressure (SLP) anomalies over winter 2014/15. Panel b) shows the mean SLP anomalies across all 510 historical weather analogues ensemble members. Panel c) shows the SLP anomalies of five example analogue variants where each variant is made up of three months from different historical years (i.e. Dec 2013/Jan 1989/Feb1984). Panel d) shows the average rainfall across the UK from the five example analogue variants. The analogue sampling contains ten analogue variants per ensemble member and is repeated for each of the 51 ensemble members of GloSea5 with a total of 510 analogue forecasts.**

**2.4 Generation of river flow hindcasts**

**2.4.1 Standard Ensemble Streamflow Prediction (ESP)**

A river flow hindcast archive using the standard ESP approach was compiled over the hindcast period (1993-2016) for each catchment and season. For each season in the hindcast period, three-month lead time seasonal ESP hindcasts were generated using the calibrated GR6J model forced with daily meteorological traces (rainfall and PET) from the historical observation record (four years prior to the forecast initialisation period to spin-up the hydrological model). Forecasts were then made by

310 forcing the GR6J model with meteorological traces taken from the equivalent three-month period for each year in the historical observations (1962-2015). In accordance with Harrigan et al. (2018), a leave-three years-out cross validation (L3OCV) was employed where meteorological traces from the initialisation year and the two succeeding years are excluded. This is to account for possible teleconnection persistence and avoid inflation of forecast skill. For example, for a hindcast initialised for JJA 2000, meteorological  traces of JJA from 1962…2015 were used to drive the hydrological model, excluding traces from 2000,

2001 and 2002For hindcasts initialised in 2014 and 2015, meteorological traces from 1962 and 1962, 1963 were removed respectively to satisfy the L3OCV procedure and maintain the 51-member ensemble (in the same procedure as Harrigan et al., 2018).  The mean daily streamflow for the 3-month forecast is taken as the forecast value.

**2.4.2 Historic Weather Analogues (HWA)**

A separate hindcast archive using the HWA approach was compiled over the same hindcast period as the standard ESP method

for each season. The initialisation procedure as above was followed to obtain the initial hydrological conditions for each catchment. At each forecast initialisation date, the GR6J model for each catchment was forced with daily rainfall and PET from all 510 HWA ensemble members. For example, assuming December 2013, January 1989 and February 1999 are the analogue years/months making up a particular forecast ensemble member of a given winter, the daily rainfall and temperature sequence for those analogue months/years are taken from adjusted HadUK-Grid observations and used to drive the

hydrological model after initialisation. This is repeated for each of the 510 analogue ensemble members.

Although analogues are selected to match the forecast pressure pattern, the associated rainfall and temperature needs to be adjusted to account for observed long-term trends. Suppose the forecast is initialised for December 2015 and the first month of the analogue forecast takes historical rainfall observations from December 1990, the analogue rainfall should account for

the long-term observed increasing trend in December rainfall between 1990 and 2015, so the analogue rainfall is appropriate for the specific forecast year. In this specific example, an estimate of the long-term trend (mm over 25 years between 2015 and 1990) relating to factors other than the NAO (see Stringer et al.in preparation) is used to modify analogue rainfall amounts. This changes the 1990 rainfall sequence by distributing the contribution from the trend according to the proportion of monthly rainfall falling on each day (i.e. if a day has 25% of the monthly rainfall, 25% of the rainfall due to the trend is applied to it).

In the hindcast experiment, it is possible for analogue years to be selected from "future" years after the year of forecast

initialisation and the long-term trend is removed in such a case. The long-term trend in rainfall is positive for all months apart from April. Trends are stronger in autumn and winter (both >0.0045mm/day/year) compared to weaker trends in spring (0.001 mm/day/year) and summer (0.003 mm/day/year). For temperature, the estimated trend is simply added to or removed from the daily temperature sequence prior to the calculation of PET. As expected, the long-term trend in temperature is positive for

all months and the highest for summer (average +0.024°C/day/year) followed by spring (0.021°C/day/year), autumn (0.017°C/day/year) and winter (0.013°C/day/year). Hydrological hindcasts are always initialised from the start of each season.

## 2.5 Skill Scores

Forecast skill is evaluated by using statistical skill scores which measure the quality of the hydrological forecast against observations relative to the accuracy of a benchmark forecast against observations. The benchmark forecast is taken as either

a probabilistic climatology forecast or the standard ESP forecasts. Simulated river flow over each forecast period driven by observed meteorological data (i.e. the baseline retrospective simulation) for each catchment is used as proxy observations from which skill scores are calculated rather than a direct comparison against observed river flows. This is a common approach to isolate the effect of improvement in meteorological forcing data rather than hydrological model error or biases (Harrigan et al., 2018; Pappenberger et al., 2015; Wood et al., 2016). Our use of the "retrospective simulation" also enables direct

comparison with previous hindcast skill assessment of the standard ESP approach at the same UK catchments by Harrigan et al., (2018). The Continuous Ranked Probability Score (CRPS) and the Ranked Probability Score (RPS) and their equivalent skill scores (CRPSS and RPSS) are chosen to evaluate the skill of the hindcasts. CRPS and RPS reward reliable (statistically consistent with observations) and sharp (confident, concentrated) forecasts, and indirectly also captures discrimination (the ability to distinguish between different outcomes). Additionally, CRPS/RPS are strongly bias-sensitive (Leutbecher and

Haiden, 2021), further supporting the choice to assess skill score of the meteorological forecast method against retrospective river flow simulation rather than observed river flows to minimise potential to over-penalise the bias. Forecast skill is defined as neutral if CRPSS values are between ±0.05 and skilful if CRPSS values are above 0.05, in accordance with Harrigan et al. (2018).

An ensemble size correction was applied given the differences between the number of members in the standard ESP (51 members) and the HWA (510 members) methods following Ferro et al. (2008). The ESP and HWA forecasts are evaluated against a probabilistic benchmark climatology created following the approach set out in Harrigan et al. (2018) from the climatological distribution of observed simulated river flows over 1965–2015 for each forecast period. While the CRPSS is calculated based on absolute deviation of the forecast value with the observations, the RPSS assess whether the forecast values

are placed within the same category of river flow values as in the simulated observations. The river flow percentile thresholds used within the operational UKHO are detailed in Table S1 and were selected to categorize observed and forecast values into five classes (low to high flows) based on the distribution of historical observed simulated river flows. The Relative Operator Characteristic (ROC) score is further calculated to explore whether each forecast method correctly predicts the occurrence of

high or low flow events. Thresholds for high and low flows are defined based on the upper and lower tercile of observed simulated flows for each catchment, following the approach taken in Donegan et al. (2021). The "easyVerification" R package was used to calculate all skill scores (MeteoSwiss, 2017). For CRPSS and RPSS, a value of 1 indicates perfect skill against the benchmark and a negative value indicates greater skill in the benchmark compared to the forecast method. For the ROC score, the area under the curve (AUC) has a maximum value of 1, indicating that all ensemble members correctly predict the occurrence for a given high or low flow event.

## 3 Results

The results are presented in the following order. First, the spatial distribution of skill (measured by CRPSS and RPSS) are presented for both the standard ESP and the HWA approach with a comparison of skill between the two approaches. Second, the ability of both methods in discriminating high and low flow events are evaluated. Third, a specific focus on hindcast for the winter season is presented with several case studies related to the link between modes of climate variability and river flow predictability.

### 3.1 Spatial distribution of skill

Figure 3 shows the spatial distribution of skill for the standard ESP and the HWA forecasts. The ESP forecasts are most skilful compared to the benchmark climatology for winter (58% of catchments with CRPSS > 0.05), followed by autumn (53%), summer (30%) and spring (29%). For the HWA forecasts, however, this ordering is slightly modified, with most skill (compared to the benchmark climatology) seen in winter (73%), followed by autumn (67%), spring (50%) and summer (30%). UK-wide mean ESP skill against benchmark climatology is highest in winter (0.09), followed by autumn (both 0.06) and lowest in summer (0.03) and spring (0.02). For HWA, UK-wide mean skill is also highest in winter (0.13), followed by autumn (0.09), spring (0.08) and summer (0.03). Figure S2 shows skill scores aggregated across UK regions for both the ESP and HWA methods against benchmark climatology across the four seasons. As shown in Figure 3, there is a distinct spatial variation in skill during summer, with skilful forecasts for catchments in the south and east UK but neutral or negative skill elsewhere, particularly in northern England and Scotland. On a regional basis, the most skilful region is the Anglian region in winter and the least skilful is Northern Ireland (NI) in summer for both ESP and HWA. The Anglian and South East regions remain the most skilful regions across all seasons for both methods and are the only two regions showing moderate skill on average in summer. There is substantial variation in CRPSS values for catchments within each hydroclimate region, for example CRPSS values for catchments in West Scotland range from -0.05 to 0.1 for the standard ESP method and from 0.06 to 0.2 for the HWA method. For a contrasting region in the South East, CRPSS values range from -0.04 to 0.5 for ESP and 0.01 to 0.5 for HWA. The equivalent skill scores calculated using observed river flows (i.e. instead of retrospective simulated river flows) shows a very similar spatial distribution with a clear improvement in winter flow predictability across northern catchments using the HWA approach, with low skill in summer months away from the southeast for both methods (Figure S3)

Skill scores calculated by comparing HWA forecasts against the standard ESP method show that the greatest improvement in skill with the HWA method is seen in winter, particularly for upland catchments in northwest NI, North West England and Scotland (Figure 4). The spatial distribution of skill is comparable between CRPSS and RPSS. There is also some improvement in spring compared to ESP, particularly for catchments in North East England and Scotland (WS and ES), where 80% (24 out of 30) and 73% (57 out of 78) of catchments had a positive CRPSS respectively. In other regions, skill for ESP and HWA is generally comparable with little spatial coherence except for some catchments in Southwest England and South Wales (SWESW) and Northern Ireland (NI) which shows a decrease in skill. Skill in most catchments, particularly in northern and western areas is reduced compared to ESP in the summer with the exception of some in central and southern England where skill is comparable to ESP. It should be noted that away from central and southern England, catchments generally show neutral (CRPSS ±0.05) or poor skill for both HWA and ESP methods for the summer.

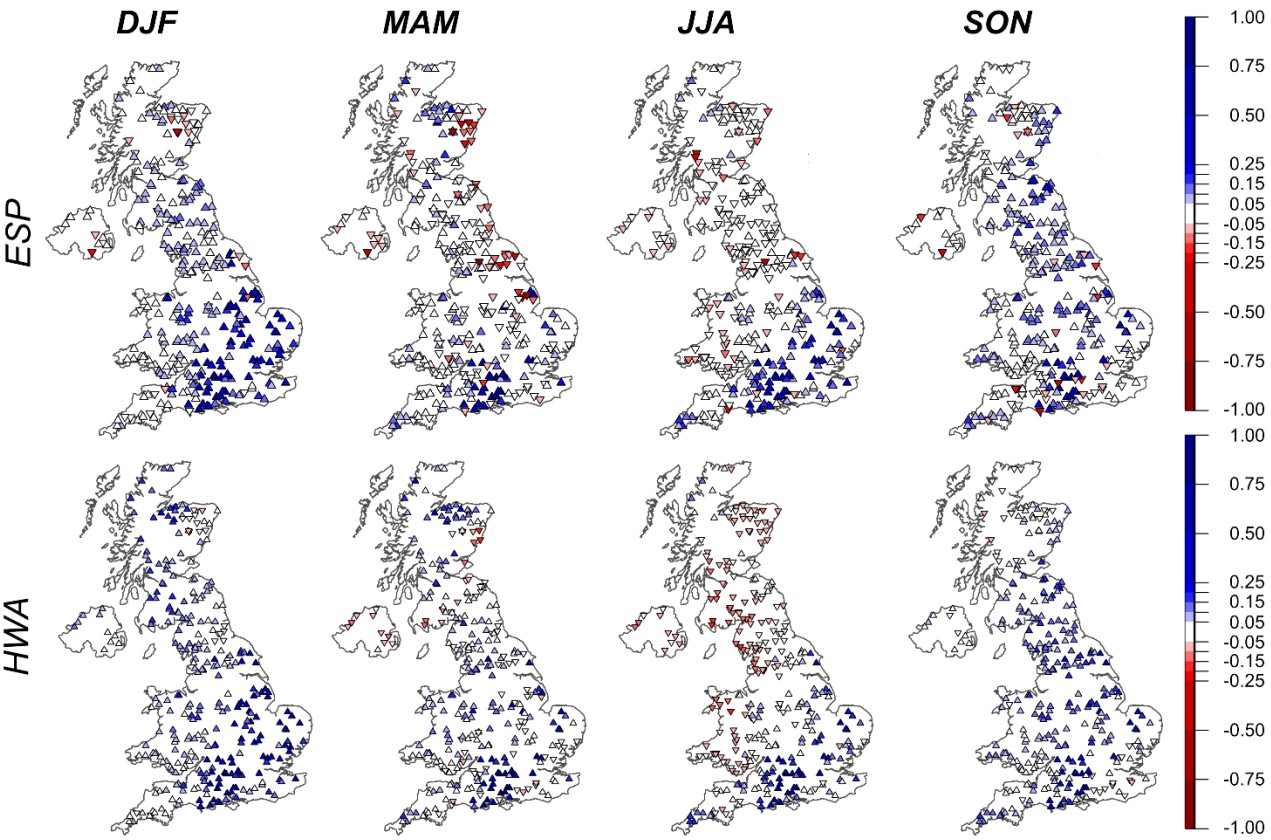

**Figure 3: Probabilistic hindcast skill for the ESP (top) and HWA (bottom) methods across the hindcast period (1993-2016) for 314 UK catchments. The metric used is the CRPSS, and is calculated for the hindcast period by comparing HWA and ESP with benchmark climatology (retrospective simulation) per season. Blue colours indicate the historic weather analogues method has higher skill than the benchmark climatology (red colours show the historic weather analogues method is worse than climatology). White colours indicate neutrally skilful forecasts. The direction of the symbol indicates the sign of the respective skill score.**

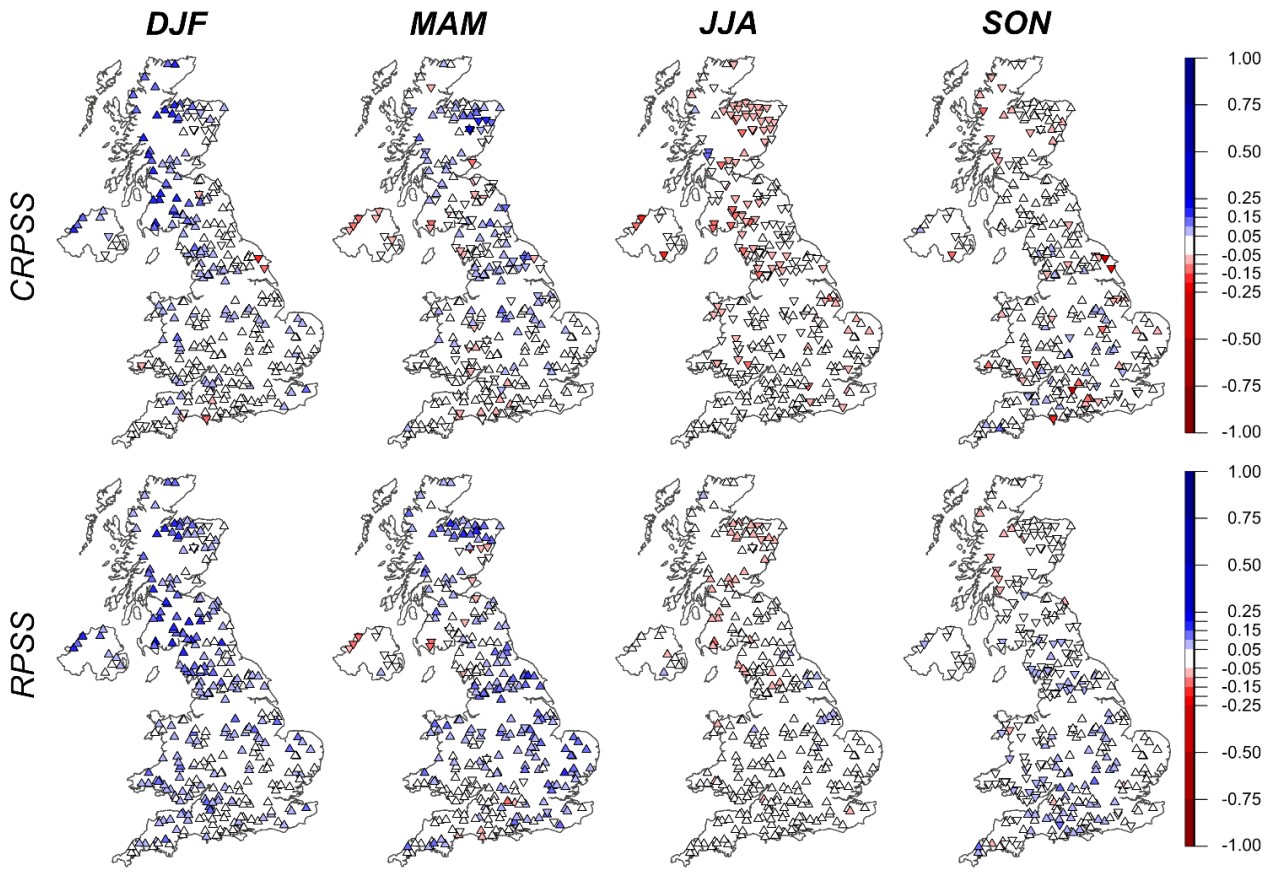

**Figure 4: Difference in skill levels between the HWA and ESP methods. HWA CRPSS (top) and RPSS (bottom) are shown for the hindcast period (1993-2016) for each season across 314 UK catchments. Blue colours indicate the HWA method is better than the Ensemble Streamflow Prediction (ESP), red colours show the HWA method is worse than ESP. White colours indicate comparative skill between the two methods. The direction of the symbol indicates whether the HWA skill is better (upward arrow) or worse (downward arrow) than benchmark climatology according to CRPSS and RPSS.**

Figure 5 shows the proportion of catchment within each hydroclimate region with positive CRPSS and RPSS for HWA against a baseline of ESP for all seasons. Generally, RPSS shows improvement for more catchments compared to CRPSS. The skill scores show a clear improvement in winter for catchments in northern England, NI and Scotland. For winter, at the national scale, 45% of catchments have a positive CRPSS (64% with a positive RPSS), indicating improvement in skill from the HWA method compared to ESP. All the catchments in Western Scotland (35) and the North-west England and North Wales region (NWESW) (35) saw an improvement in skill with the HWA method when assessed using CRPSS. In spring, the proportion of catchments with an improvement in forecast skill compared to ESP are less notable than in winter but there is still a notable

improvement nationally (including across Scottish and northwest regions) when assessed using both CRPSS and RPSS. 7 of the 9 regions show an improvement in forecast skill for over 50% of catchments in spring when considering CRPSS. Forecast improvement in autumn is seen for catchments in North East England and NWENW and regions in the southeast although the magnitude of that improvement is more modest compared to winter and spring (as seen in higher CRPSS values calculated by comparing HWA against ESP in Figure 4). Generally, summer forecast skill using HWA is comparable or worse than the standard ESP for the majority of catchments in each region. When assessed using CRPSS, improvements were only found for a handful of catchments in all regions with no regions registering improvements in more than 50% of catchments.

The skill scores during winter for individual catchments within the hydroclimate regions show a large degree of sub-regional variation in forecast skill (Figure 6 for the West Scotland and Anglian regions and Figure S4 for other hydroclimate regions). For example, even within the Anglian region where winter flow forecasts averaged across the region exhibit good skill, some catchments (e.g. ID: 36010 Bumpstead Brook) show limited skill while others show very high skill (e.g. 34014 Wensum). The improvement in winter forecast skill is negatively correlated with the baseflow index (BFI) (r=-0.44, *p-value < 0.05*) and positively correlated with standardised annual average rainfall (r=0.51, *p-value < 0.05*) and the catchment wetness index (r=0.64, *p-value < 0.05*) (Figure S5). The relationship shows that catchments with the greatest improvement in winter skill tend to be fast-responding catchments with a low baseflow contribution to river flows and are often wet catchments located in upland regions with higher annual average rainfall.

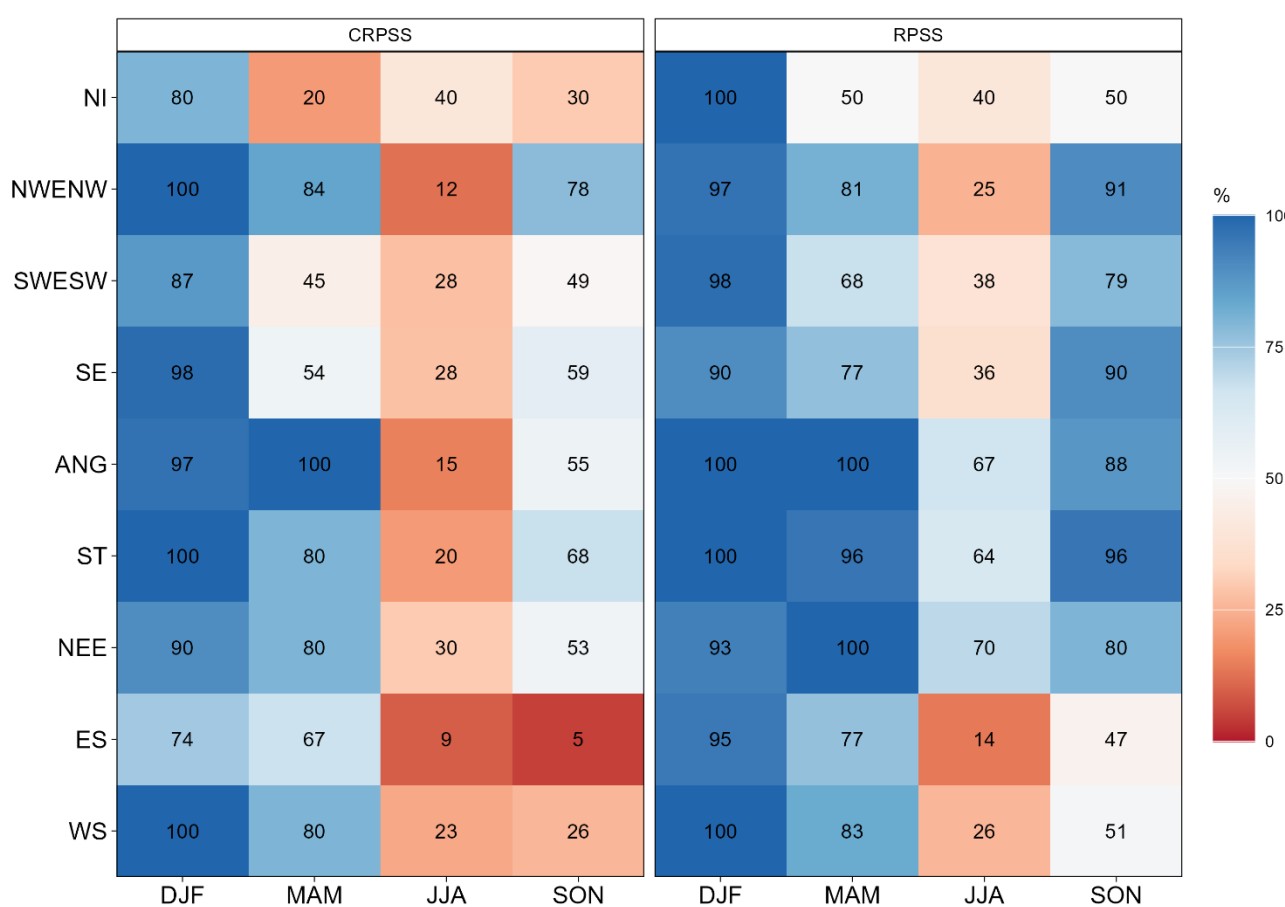

**Figure 5: Percentage of catchments with positive CRPSS (left) RPSS (right) values calculated from comparing the HWA forecasts against the standard ESP forecasts for each season and each hydroclimate region. The percentage number is shown for each region and season. Regions with more than 50% of catchments showing improvement are shaded in blue colours whereas regions with less than 50% of catchments showing improvement is shaded by red colours.**

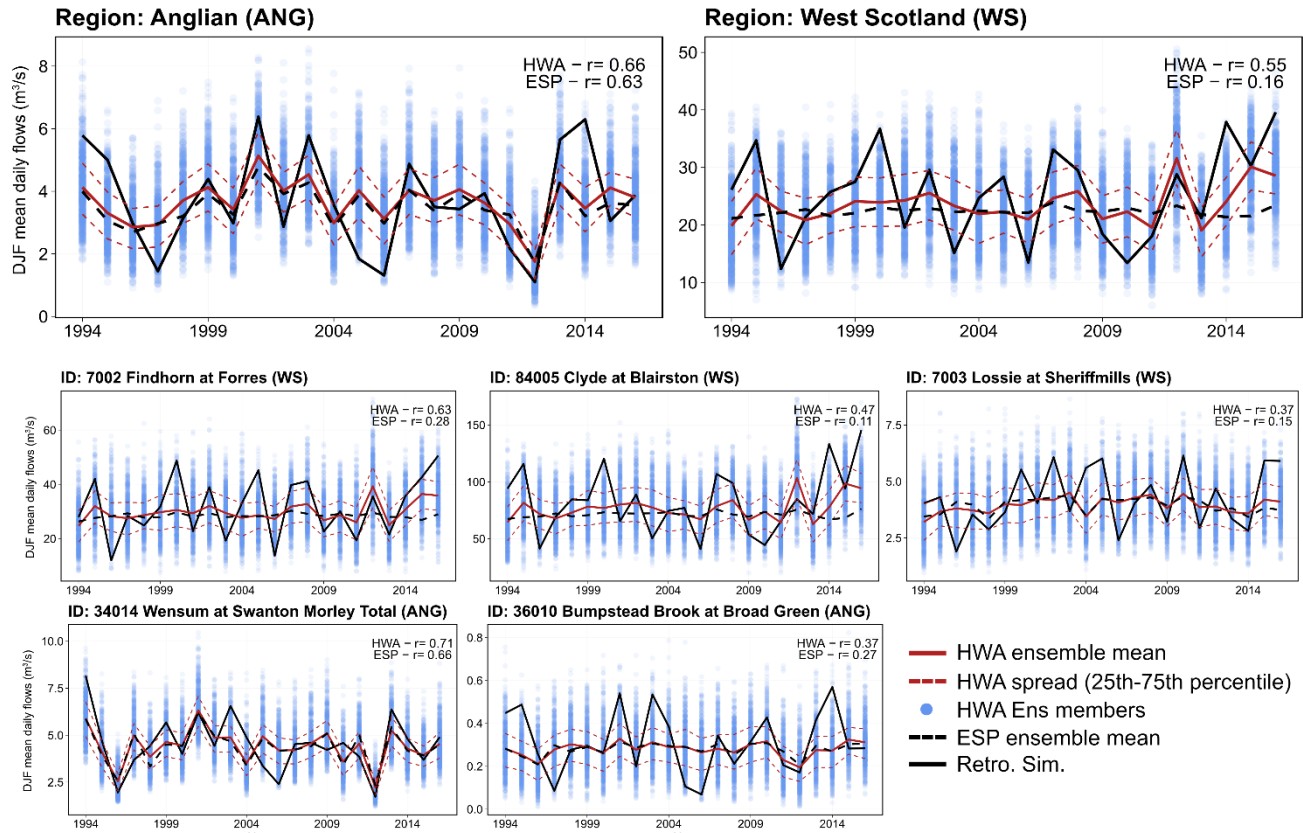

Figure 6: December-January-February (DJF) mean daily simulated flows (m³/s) averaged across catchments in two contrasting regions (West Scotland and Anglian) and for five individual example catchments with varying skill in each region. Blue dots show the individual ensemble members from the HWA method, the red solid line shows the HWA ensemble mean, the red dotted lines show the HWA spread (25th-75th percentile), the dotted black line shows the ensemble mean of the standard ESP method and the solid black line shows retrospective simulated river flows. The correlation coefficients between the retrospective simulated river flows and the ensemble mean HWA and ESP forecasts are shown on each plot.

Figure 7 shows the skill for both methods in discriminating between events (upper or lower tercile river flows) and non-events (middle tercile river flows) for all seasons across each hydroclimate region, showing broadly consistent regional patterns with the CRPSS and RPSS results above. During winter, the HWA method results in more catchments with good skill (ROC score > 0.6) in discriminating both low and high flows across all regions compared to the standard ESP where good skill was only seen for regions in the southeast. ROC scores for catchments in southern England (i.e. SE and ANG regions) are comparable between the two methods for both high and low flow events with clear improvement in skill for all other regions, specifically those in Northeast England and Scotland (WS and ES). Across all considered catchments, skilful discrimination of low flow events in winter is achieved for 41% of catchments using ESP and 88% of catchments with the HWA method, while skilful discrimination of high flow in winter is achieved for 35% of catchments with ESP and 72% with HWA. In spring, the HWA

method generally shows skilful discrimination of high and low flow events in most regions, except for low flows in NI where skill deteriorated compared to ESP. ROC scores for other seasons are similar between ESP and HWA with some regions showing slight deterioration. Skill is particularly low across summer months for both ESP and HWA, highlighting that skill in discriminating low and high flows in summer remains a particularly notable challenge. Some regions saw a reduction in skill in summer with HWA compared to ESP (e.g. NI, ES and WS for high flows).

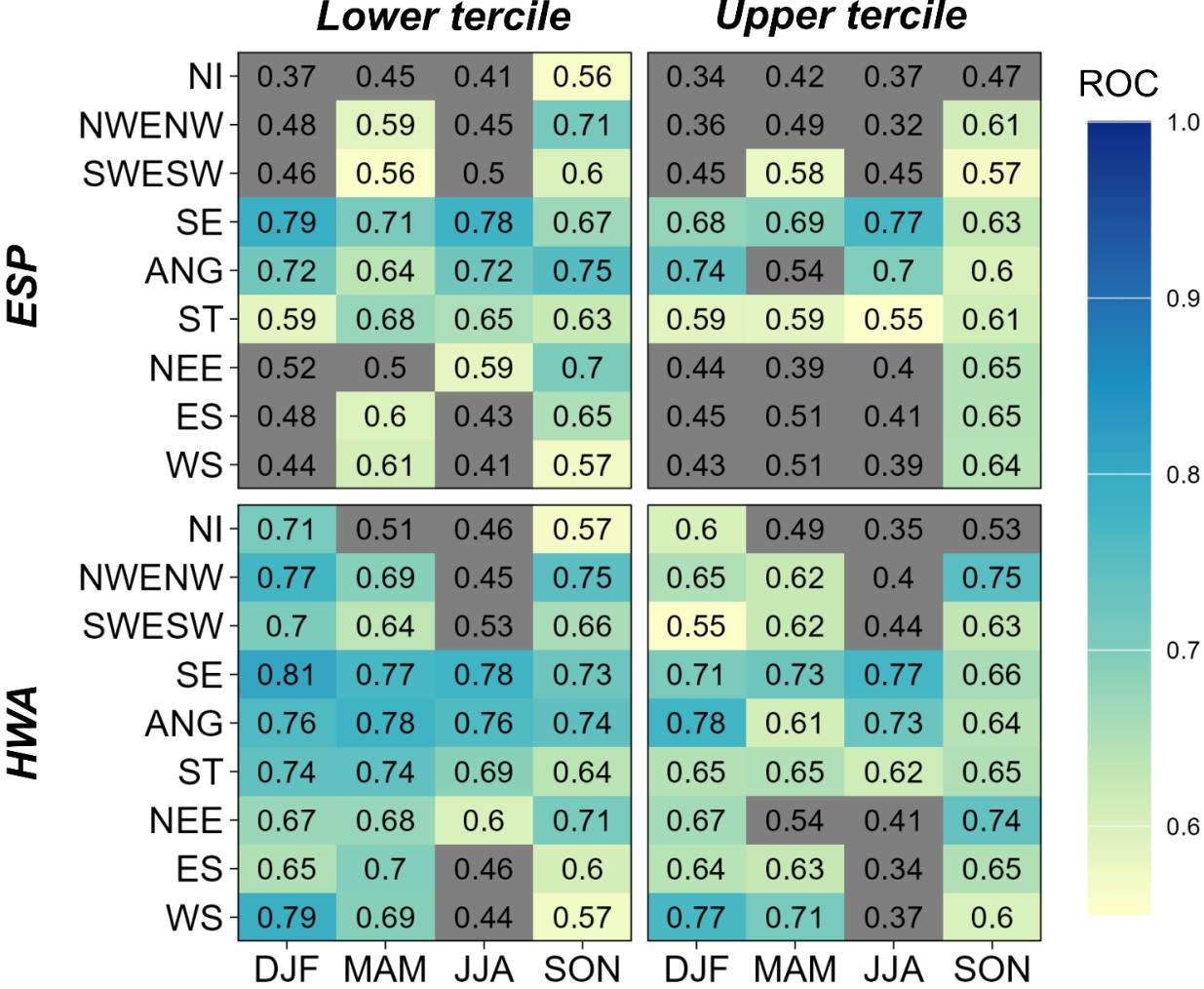

**Figure 7: ROC scores averaged across each hydroclimate region for the standard ESP method (top) and the HWA method (bottom) for each season and for lower tercile (low flows) (left) and upper tercile (high flows) (right). Cells with ROC scores < 0.6 are greyed out.**

**3.2 Winter flow predictability**

Winter saw the highest number of catchments registering improved seasonal flow predictability against the standard ESP method, particularly for upland, fast-responding catchments in Northern England and Scotland. For other seasons, studies have shown that total rainfall variability is less well explained by the leading modes of climate variability and the influence of global weather patterns on UK weather also tends to smaller. Hence, this section aims to further understand predictability for winter river flows in more detail by linking river flow predictability to large scale atmospheric drivers. Figure 8 shows the proportion

of river flow forecasts in the correct category compared to that in the retrospective simulation calculations for both ESP and HWA averaged across West Scotland for each winter within the hindcast period. There is an increase in the percentage of correct forecasts with the HWA method in 16 of the 23 winters. Forecast performance improvement is more apparent for wetter than average winters, with a higher proportion of forecasts in the correct category for HWA compared to ESP for 11 of the 13 wetter than average winters. There is a mixed picture for drier than average winters, for example the HWA method showing

better forecast performance during winter 2010 but poorer performance in winters 1996 and 2006, all of which coincide with notable national hydrological drought episodes, but a longer hindcast period and larger ensemble is required for a more robust assessment.

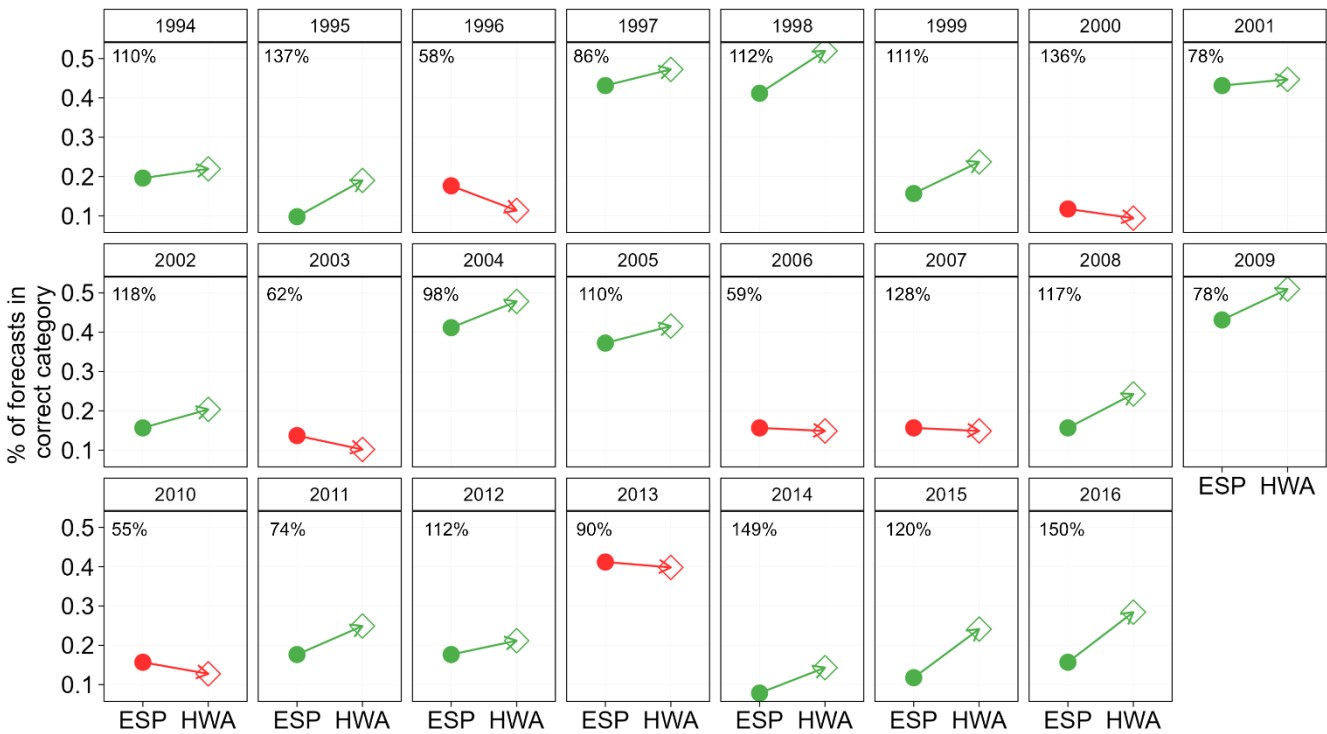

Figure 8: Change in the fraction of DJF forecasts in the correct category (compared to the retrospective simulation) averaged across catchments in Scotland. Dots on the left show the standard ESP method, diamonds on the right show the HWA method. Green arrows indicate years with an improvement in the forecasts with the HWA method, red arrows indicate a decrease in forecast performance. Rainfall as a percentage of average for each winter is shown on each panel. The year in the title refers to the year of the January and February.

Two case studies of winters with strong NAO signals are selected to further explore winter river flow predictability. Winter 1994/95 had the strongest NAO+ signal in the hindcast period while winter 2009/10 was in a strong NAO- phase. Figure 9 shows the probabilistic ESP and HWA hindcasts and the retrospective simulation river flows for both winters, visualised in the same format as the operational UK Hydrological Outlook and interactive online portal (https://ukho.ceh.ac.uk). Winter 1994/95 remains the third wettest UK winter since 1836, only surpassed by winters 2013/14 and 2015/16. Numerous catchments exceeded their mean winter river flows record at the time but it was notably followed by a major drought after widespread rainfall deficits from spring 1995 onwards (Institute of Hydrology and British Geological Survey, 1996). Hindcasts for winter 1994/95 using the standard ESP method predicts normal to below normal river flows across southern England with an increase in likelihood for below-normal to low river flows elsewhere. Hindcasts using the HWA method instead indicates a reversal, showing a heightened likelihood of above-normal to high river flows for many catchments across northern and western Britain. By construction, the normal category will often be the most likely in any forecast given the percentiles chosen.

Nevertheless, the HWA method gave warning of the increased risk of the high flows for this winter, unlike the ESP method which suggested the risk was lower than normal.

Winter 2009/10, the third coldest winter for Scotland since 1836, was characterised by a strongly negative NAO- phase and prolonged dry conditions with a marked east-west spatial contrast for rainfall (Prior and Kendon, 2011). Rainfall was notably below average across western UK with some areas receiving less than half the average (Kendon et al., 2013). The winter was a precursor to the multi-year 2010-12 drought, which began with widespread below normal winter river flows across western Britain (Marsh et al., 2013). Both the ESP and HWA hindcasts for river flows indicate a higher likelihood of below-normal to

low river flows across the UK, agreeing well with what was observed. While the spatial pattern from both methods were broadly similar, the likelihood of low river flows is heightened in the HWA method for most catchments compared to ESP. This further illustrates results from the skill scores showing the improved ability of the HWA method to discriminate between both high and low flow events.

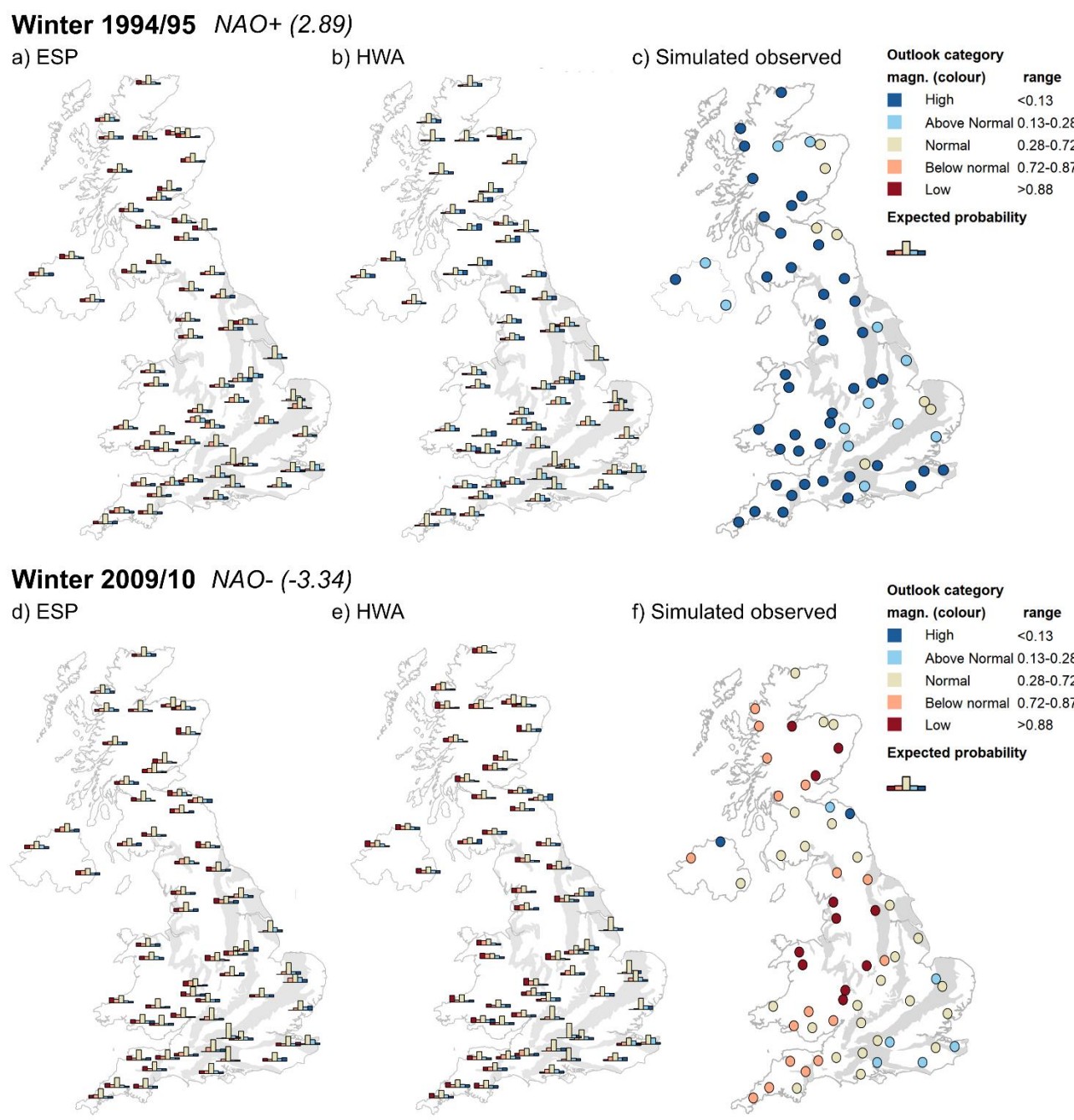

**Figure 9: Case studies from the hindcast experiment for DJF 1994/95 (top) and 2009/10 (bottom). ESP hindcasts are shown in the left panels, with HWA hindcasts in the central panels, and the river flow category for retrospective simulated river flows flows in the rightmost panels. Note the visualisation style is the same as the operational UK Hydrological Outlook and in the interactive online portal. The colour scheme has been adopted after extensive stakeholder consultation.**

Winter river flows at catchments in North West UK, and particularly Scotland strongly reflect rainfall variability given their

responsive nature. As winter NAO is a major mode of climate variability associated with winter UK rainfall, the ability of the

HWA forecasts to consider predicted dynamical signals substantially improves winter river flow predictability. This contrasts

with slower-responding catchments in other parts of the UK where initial hydrological conditions play a stronger role. The

HWA approach does not sample for analogues using the NAO index directly, but instead selects analogues based on the spatial

MSLP pattern. As the NAO is a statistical description of atmospheric circulation, North Atlantic jet stream characteristics are

a more direct characterisation of atmospheric circulation (e.g. winter jet stream intensity or latitude). Figure 10 shows that

winter rainfall across Scotland is highly correlated with the NAO index (r = 0.80) and the North Atlantic jet stream intensity

(r = 0.87). Winters with strong positive NAO are also often associated with strong jet stream intensity. Within the hindcast

period, six of the top ten years with the greatest improvement in the HWA river flow forecasts relative to ESP (i.e. higher %

of ensemble members in the correct category) are in a positive NAO phase (NAOI > 1) and are associated with a strong jet

stream intensity. The top three years with greatest improvement are 1995/96, 2015/16 and 2013/14. Given the relatively short

hindcast period and the large atmospheric circulation variability, a longer hindcast period is required to understand whether

there is a systematic and statistically significant differences in forecast improvement conditional on diverging NAO phases.

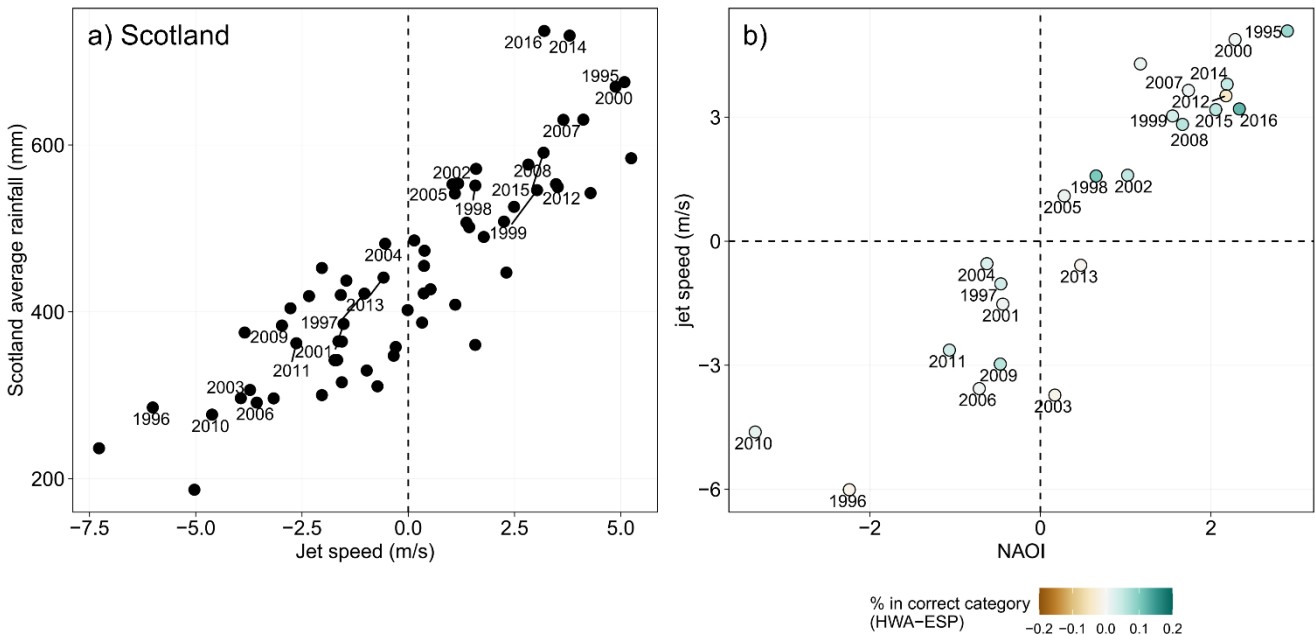

**Figure 10: Linking winter river flow predictability to atmospheric circulation patterns. a) Scotland average rainfall against mean winter North Atlantic jet speed (m/s) for 1956-2018, with the years in the hindcast period highlighted. Panel b) shows jet speed (m/s) against the North Atlantic Oscillation (NAO) index, with the dots coloured by the percentage of HWA forecasts in the correct category as the retrospective simulation minus the percentage of ESP forecasts in the correct category as the retrospective simulation**
**averaged across catchments in Scotland for all years in the hindcast period - positive values indicate a higher percentage of forecasts in the correct category in HWA compared to ESP.**

## 4 Discussion

This study completed a rigorous assessment of forecast skill using both the standard ESP approach and a new Historic Weather Analogues (HWA) approach over a common hindcast period. Overall, we find that the HWA method retains much of the forecast skill of the standard ESP but is a clear improvement in winter months with modest improvements in spring. The degree of skill improvement is dependent on the season and on climatic and physical properties of individual catchments.

### 4.1 When and where is HWA skilful?

The HWA forecasts are skilful relative to climatology for most catchments in the Anglian, Severn Trent and South East regions for all seasons. In the winter, the HWA forecasts are also skilful for catchments in in North West England and Scotland (CRPSS > 0.05 for 73% of all selected catchments). Skilful forecasts are also possible for some catchments in spring and across North West and North East England in autumn. In the summer, the HWA forecasts exhibit neutral to negative skill for catchments across North West England and Scotland and skilful forecasts remain possible only for catchments in the south and east. The source of forecast predictability over catchments in the south and east comes from the persistence of initial hydrological conditions as this region contains mainly groundwater dominated catchments with high catchment storage capacities, contrasting with relatively lower skill for fast responding catchments where initial hydrological memory is quickly lost (Harrigan et al., 2018). Catchments in the southeast also yield skilful persistence forecasts as shown by Svensson et al. (2016). The importance of initial hydrological conditions in achieving skilful hydrological forecasts in the UK is in line with conclusions in previous hindcast experiments using ESP or modified ESP methodologies (Svensson, 2016; Svensson et al., 2015). Harrigan et al. (2018) showed that the total catchment storage (defined as the sum of storage components within the GR4J hydrological model) is strongly correlated with ESP skill across UK catchments and that skill declines at a much slower rate for catchments with high storage capacity (e.g. skilful forecasts can be made at seasonal or longer lead times for catchments with high storage).

Investigating the predictability of the winter NAO in various forecasting systems (including the GloSea model), Baker et al. (2018) found that they are able to capture the global physical climate drivers and teleconnection pathways and skilfully forecast winter NAO across Europe at seasonal or longer lead times. The HWA approach makes use of both the influences of initial hydrological conditions and the predictability of atmospheric circulation patterns. The ability to consider atmospheric circulation predictability and their influence on regional UK rainfall is especially important for North West England and West Scotland. Catchment storage is often limited in these regions and rainfall is directly translated into river flows with catchments often exhibiting a "flashy" flow regime that is highly correlated with rainfall variability and atmospheric circulation indices such as the winter NAO. While HWA method considers predictability of atmospheric circulation patterns in all seasons, the results show higher skill during winter. This is due to higher forecast skill for winter circulation patterns, thus providing a

sound justification to amplify the predicted ensemble mean dynamical signals (e.g. winter NAO) to account for the signal-to-noise problem in meteorological forecasts (Eade et al., 2014; Scaife and Smith, 2018). Dynamical signals for other seasons were not amplified as forecast skill for circulation patterns during those seasons are less skilful compared to the winter months.

## 4.2 Comparison with ESP

The results used the standard ESP approach as a benchmark to assess where and when the HWA forecasts represent an improvement. Across all seasons, the HWA forecasts generally have higher skill compared to ESP across most catchments. The HWA approach represents a substantial improvement in river flow predictability in northern and western areas where high skill derives from fast catchment response to rainfall and a close association between river flow and climate variability. In contrast, with the standard ESP, high forecast skill in winter was only found for catchments in southern England where large catchment storage capacities and strong catchment memory of initial hydrolgical conditions enhances predictability at long lead times. In the summer, skilful summer flow forecasts for both HWA and ESP only remain possible across catchments in southern England and are attributed to the strong influence of initial hydrological conditions. This reflects previous findings in Harrigan et al. (2018) using standard ESP and Svensson (2016) using a data-driven flow persistence/flow analogue approach. The HWA approach, as a conditioned ESP approach, would naturally retain skill in areas with high total catchment storage, as shown by year-round skill in summer flow predictability for groundwater-dominated catchments in the south-east. However, the HWA forecasts show some deterioration in summer forecast skill across isolated catchments away from southeast England. When assessing skill individually for each method, there is little skill in the summer for either ESP or HWA away from groundwater-dominated catchments in southern England, hence catchments which showed a deterioration in skill with HWA compared to ESP are generally those which do not show skilful forecasts with ESP. For other seasons, the HWA approach shows some improvements in spring, mostly at catchments in northern England and northern Scotland and modest improvements in southeast England. The comparison of skill scores between HWA and ESP for the selected catchments in Northern Ireland in this study agrees well with Donegan et al. (2021) which applied the same set of HWA hindcasts at a wider set of NI catchments using the GR4J hydrological model, suggesting the largest improvement in skill for forecasts initialised in winter months at fast-responding catchments in northwestern NI.

The HWA method was operationalized within the UKHO during winter DJF 2023/24. Figure S6 shows the forecast issued for winter 2023/24 in December 2023. The wider winter half year 2023/24 was the wettest on record for many catchments across England & Wales (Chan et al. 2025). The ESP method suggested higher than average river flows for slow-responding catchments in south and east and a high likelihood of flows in the normal range elsewhere. The HWA method also suggested high river flows across the south and east but also a higher likelihood of above normal and high river flows for catchments in North Wales and northern England. The higher likelihood of high flows forecasted by the HWA method adds confidence to

the results found in the hindcast experiment, suggesting that improved consideration of meteorological predictability in the HWA approach is most critical in the winter months for fast-responding catchments in northern and western areas.

While the HWA method shows improvement in skill across northern and western UK in winter months, the standard ESP method remains a "tough to beat" forecasting system, and in the absence of skilful meteorological forecasts at seasonal or longer lead times, remains a computationally efficient approach for hydrological forecasts. The results provide an indication of which method may be considered more skilful when intialised in different seasons (such as greater skill for HWA forecasts in the winter) but further reinforces the fact that ESP remains a low-cost, efficient benchmark forecast method for which future

improvements, such as the direct use of skilful meteorological rainfall forecasts or forecast post-processing should be assessed against (Harrigan et al., 2018; Pappenberger et al., 2015). A particular advantage to the HWA approach that is different to past conditioned ESP approaches is the ability to explore historically unseen weather sequences from the shuffling procedure applied. This introduces greater variability in the possible rainfall and PET sequences while preserving the atmospheric circulation signal forecasted by weather forecasting systems (e.g. Beckers et al., 2016). The assumption that historic weather

sequences (i.e. rainfall and temperature) would remain static if they reoccur at the time of forecast initialisation was further addressed by applying climate trend corrections to the rainfall and temperature variables for each analogue sequence prior to their application in the hydrological model. The re-trending procedure is currently not applied to the standard ESP approach as the aim to explore the consequences if the identical temperatures and rainfall sequences in past years occur again remain important from a risk awareness perspective.

**4.3 Potential future improvements**

There are several areas for potential future work. More accurate simulation of river flows from improved process representation and parameterisation of hydrological models will enable more accurate estimation of the initial hydrological conditions at forecast initialisation. In this study, the GR6J model is selected for its relative simplicity to enable a large number of model runs efficiently. Future work could investigate hydrological model uncertainty in UK river flows forecasts. The Probability

Distributed Model (PDM) and the Hydrologiska Byråns Vattenbalansavdelning (HBV) model are examples of alternative conceptual catchment hydrological models that have previously been applied successfully at a wide range of UK catchments for both short-term forecasting and long-term climate projections. They can be used within a multi-model benchmark framework (e.g. Lane et al., 2019) to identify whether different representation of hydrological processes could lead to improved skill for catchments with different physical characteristics (such as through the ongoing Hydro-JULES model intercomparison

project for the UK - https://hydro-jules.org/uk-hydro-mip).

There are also several areas of future work to further develop methodologies within the UK Hydrological Outlook. Sources of uncertainties along the modelling chain of operational streamflow forecasting systems include hydrological model uncertainties in the representation of initial hydrological conditions and uncertainties associated with the structure and

representation of initial weather states from operational weather forecasting systems (Troin et al., 2021). Further insights into the conditions when forecasts of atmospheric circulation patterns are skilful were recently provided by Baker et al. (2024), suggesting that global teleconnections such as ENSO affects the predictability of the winter NAO with higher NAO predictability during the El Niño phase. For example, the forecast atmospheric circulation signal was relatively weak in winter 2013/14, which was a winter in the neutral ENSO phase with strong stratospheric influences (Huntingford et al., 2014). Conversely, the strength of the signal for winter 2015/16 was larger and it was part of one of the strongest El Niño events on record. Improvement in river flow predictability was comparatively large for winter 2014/15 than for winter 2013/14 (i.e. Figure 8). A more systematic influence of ENSO on NAO and thus winter river flow predictability may be confirmed with future work employing a longer hindcast period. Further refinement of the analogue selection approach could consider predictability of other modes of variability which may provide justification for the amplification of dynamical signals in other seasons. Improvements in the predictability of the East Atlantic pattern, which exhibits strong influence on rainfall variability, particularly across southern Britain (West et al., 2019), could contribute to further advances in national summer river flow predictability. Potential improvements in summer flow predictability can also be enabled with better understanding of large-scale atmospheric teleconnections. For example, Chevuturi et al. (2025) recently demonstrated that summer flow predictability across northern and western UK is linked to variations in North Atlantic sea surface temperatures and resulting atmospheric teleconnection pathways with signs of long-range predictability (~1.5 years ahead).

Applying post-processing techniques such as bias correction and multi-model blending can also substantially improve the reliability and performance of hydrological forecasts (Arsenault et al., 2015; Chevuturi et al., 2023; Matthews et al., 2022; Troin et al., 2021). For example, Chevuturi et al. (2023) recently tested several approaches to blend hydrological forecasts from multiple global land surface and hydrological models. They found that applying a weighted average of multi-model forecasts based on model performance in conjunction with bias correction yields the greatest improvement in forecast skill and can be a suitable method to communicate forecasts to end-users. Tanguy et al. (2024) used the UKHO as a case study and showed that a quantile mapping bias correction post-processing technique is a computationally efficient and low-cost way to improve the skill of hydrological forecasts by correcting for systematic biases in hydrological model simulations. The authors also showed that data assimilation can further improve ESP forecasts by adjusting the internal hydrological model states at forecast initialisation, which may be beneficial at catchments with high storage as a more accurate estimation of initial hydrological conditions is crucial. However, data assimilation is computationally demanding and requires the availability of live, near real- time observations, which remains a major drawback to operational use. Further work is on-going to blend the catchment-based results presented in this study from different forecasting methods with those made using gridded hydrological models. An eventual aim is to operationalise a multi-method and multi-model blended forecasting system that either chooses the most effective method/model for each lead time/season (e.g. applying the HWA method in place of standard ESP at fast-responding catchments) or to weigh results based on the performance and reliability of different models/methods over a hindcast period (Tanguy et al., 2024).

The current set up of the seasonal hindcast archive cannot be used for shorter lead times as the circulation analogues were chosen to match the seasonal mean forecast MSLP pattern. However, sub-seasonal (i.e. monthly) forecasts using the HWA method are also made within the UKHO using an analogous system where the forecast month are split into three 10-day segments and the best match of the forecast circulation pattern is chosen from the 10-day segments instead of the seasonal mean. The application of HWA forecasts at even shorter lead times (e.g. sub-monthly and weekly timescales) are currently in

development (Rhodes-Smith and Bell, 2024). Hybrid forecasts coupling statistical or machine learning techniques with dynamical climate model forecasts further shows promising opportunities to enhance hydrological forecasts ranging from sub-seasonal to decadal timescales (Golian et al., 2022; Slater et al., 2023; Slater and Villarini, 2018). For example, sub-selecting hindcasts from decadal climate simulations based on their representation of atmospheric circulation variability (e.g. winter NAO) coupled with a statistical flood model for the UK showed skilful indication of flood rich or flood poor decades,

particularly for fast-responding catchments in northwest UK (Moulds et al., 2023). This further echo the results found in this study, illustrating the benefits of incorporating climate information for forecasting river flows at catchments where meteorological variability is a direct driver of flow variability.

## 6 Conclusions

The Historic Weather Analogues (HWA) method builds on the existing suite of forecasting methods within the UK Hydrological Outlook and aims to leverage the improved predictability of atmospheric circulation patterns to improve the skill of seasonal river flow forecasts across the UK. This study uses both climatology and the standard Ensemble Streamflow Prediction (ESP) method as benchmarks to assess river flow forecast skill of the new method at catchments across the UK. The HWA forecasts represent a clear improvement when compared to the standard ESP, notably so for winter river flow

predictability, particularly for catchments in North West England and Scotland. This is compared with ESP where skilful winter river flow forecasts were only possible for catchments in South East England, where initial hydrological conditions related to groundwater storage provides high seasonal predictability. Catchments with the greatest improvement in winter river flow forecast skill tend to be upland, fast responding catchments with limited catchment storage and where river flow variability is strongly tied with climate variability. Results also show that the HWA forecasts have greater skill in

discriminating both high and low flow events in the winter compared to ESP. The improved winter river flow predictability derives from the predictability of winter NAO (and the ability of the HWA forecasts to match the predicted atmospheric circulation pattern), apparently with more notable improvements in the forecasts of high river flows during wet winters in a positive NAO phase although a longer hindcast period is required to confirm this.

Skilful prediction of summer river flows, particularly for catchments away from South East England, should remain an outstanding research priority. In the summer, both ESP and HWA are skilful only for slow-responding catchments in southern

England with the HWA method showing similar or slight reduction in skill elsewhere.Forecasts for other seasons (spring and autumn) suggest modest improvements with larger improvements against ESP during spring, with no significant decrease in skill compared to ESP. Future work such as post-processing of river flow forecasts through bias correction and data assimilation may further improve forecast skill at catchments which the HWA forecasts already exhibit strong improvements. The amplification of other dynamical signals, such as the East Atlantic pattern, and the improved knowledge of global teleconnections driving UK rainfall variability could ultimately improve forecasts of river flows across all seasons.

This study demonstrates that the HWA method leverages climate information from dynamical weather forecasting models, leads to improvement in winter river flow forecasts in the northwest UK, and retains the forecast skill derived from the influence of initial hydrological conditions, which contributes to high forecast skill at South East England catchments for all seasons. For spring, summer and autumn, the standard ESP approach remains a "tough to beat" river flow forecasting system against which future improvements can be assessed.

**Funding Acknowledgements**

The UK Hydrological Outlook is principally funded through the UKCEH National Capability for UK Challenges Programme [NE/Y006208/1], with scientific developments supported through the HydroJULES programme [NE/S017380/1]. Underlying research and methodological development have also been supported by the Natural Environment Research Council (NERC) Climate change in the Arctic–North Atlantic Region and Impacts on the UK (CANARI) project [NE/W004984/1].

**Author contributions**

All authors contributed to the conceptualisation and design of the research, as well as to the preparation and revision of the manuscript. WC: conceptualisation, formal analysis, visualisation, writing (original draft), writing (review and editing). KFC: conceptualisation, methodology, funding acquisition, project administration, supervision, writing (review and editing). MT: conceptualisation, methodology, writing (review and editing). EM: data curation, methodology, writing (review and editing). BB: data curation, methodology, writing (review and editing). NS: conceptualisation, data curation, methodology, writing (review and editing). JK: conceptualisation, data curation, methodology, writing (review and editing). JH: conceptualisation, funding acquisition, project administration, supervision, writing (review and editing).

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
