# Peer review of "UK Hydrological Outlook using Historic Weather Analogues"

_EGUsphere, 2025_

## Author Comment (AC1)

**UK Hydrological Outlook using Historic Weather Analogues**

**Response to reviewers**

**Reviewer #1**

Thank you for this well-written and timely manuscript, which describes a new experiment in the UK national hydrological forecasting system. The method incorporates historical observations from analogue months by matching the large-scale circulation patterns, and use them as forcings to generate hydrological forecasts. The study shows improvements in seasonal forecasts skills and event categorization, particularly in winter (the rainy season). This work is both important and valuable for the hydrological forecast community.

>> We thank the reviewer for their detailed comments on our manuscript. We are glad the reviewer recognises the importance of our results and the value of the methodology for the hydrological forecast community. Please find below our response in red.

Below are some comments to further discuss the idea with the authors and improve readability:

Line 68, Consider specifying "summer NAO (SNAO)" when first time refer to it.

>> Thanks – this will be changed in the revised manuscript.

Line 86, section 1.2, The section mentions four forecast categories, but the introduction states there are "three strands." Please clarify this. And the methods in the first category might be better to conclude as "descriptive forecasts" to distinguish them from ensemble-based approaches that come later.

>> Thank you for noticing our error. We will make clear in our revised manuscript the four strands. We will also adopt the reviewer's suggestion and to distinguish between "deterministic forecasts" with "ensemble-based approaches", also addressing reviewer 2's suggestion of aligning this section with standard forecasting terminology.

Line 144, Are analogue months considered independently (i.e., monthly NAO indices)? Have you tested using moving-window averages for NAO to account for variability in selecting analogues?

>> Analogue months are considered independently. In the winter seasonal forecasts (i.e. DJF), the monthly NAO values are accounted for in the analogue selection procedure. However, when matching MSLP patterns, only the seasonal mean MSLP pattern I used for analogue selection based on spatial similarity of the forecast MSLP pattern. We will clarify this in the revised manuscript.

Line 220, Could you clarify why 17 ensemble members were chosen here? A flow chart illustrating the selection process would be helpful.

>> There are currently 17 hindcast members available for each of the three initialisation dates (i.e. giving 51 ensemble members per season). The resulting 51 ensemble members is broadly similar in size to the operational forecast so is considered a fair reflection of the operational forecast skill. We will more clearly point readers to Stringer et al., (2020) which provides the details of the hindcast ensemble and analogue selection methodology.

Line 235, For analogue season selection, have you plotted rainfall patterns for an example season to assess consistency among analogue months? It would be interesting to see such visualization (e.g., a map or time series).

>> Thank you for the suggestion. We would include a time series of the hindcast ensemble mean rainfall and observed mean rainfall for the different seasons in the revised manuscript.

We note that our companion paper (Rhodes-Smith et al., in review - https://doi.org/10.5194/egusphere-2025-2506) already includes a figure showing correlation of hindcast ensemble mean rainfall and observed rainfall across Great Britain (figure 2 in their paper).

Line 336, The text here continues analyzing results from Figure 3. But it reads like it is from Figure S1. Just specify it would help.

>> We will specify that we are writing about the results from Figure 3 here, and not Figure S1.

Line 341, What does "heterogeneity" refer to here, between the areas or between the methods? How are the numbers reflecting heterogeneity, could you explain a bit more.

>> Heterogeneity here just means large variation in CRPSS values within the different hydroclimate regions. We will rephrase to "there is substantial variation in CRPSS for catchments within each hydroclimate region".

Line 348, consider adding the catchment numbers together with the ratio, e.g. XX out of YY.

>> Thank you for the suggestion. We will add the number of catchments within each region in the revised manuscript.

Line 388, Typo: "--0.38" should likely be "-0.38." Is this value statistically significant? We will quote statistical significance in each sub-panel in the revised manuscript.

>> Thank you for spotting the typo, we will correct in the revised manuscript.

Line 420, Figure 7, This is an excellent visualization. I also noticed that for summer, both high flow events and low flow events had a drop in performance using HWA. Could this reflect challenges in low-flow forecasting? Since later in the discussion the authors mentioned summer is a future target, so maybe already mention it here while discussing the results for summer months.

>> Thank you for your comments on the figure visualisation. The reviewer is correct that summer flow predictability for summer months have dropped when using the HWA method, mainly outside of groundwater-dominated, slow-responding catchments in the south and east. We agree that this is an on-going challenge for low flow forecasting and will mention this particular challenge in the revised manuscript results section.

Line 452, Is it better to show the correct ratio for each station instead of the full distribution? Or if distributions are preferred, please just specify the reasons.

>> We have adopted to show the full distribution as this is the standard format for visualising forecast results from the UK Hydrological Outlook. The visualisation approach and the colours have been adopted after extensive stakeholder consultation and is operationalised via both the monthly Hydrological Outlook and in the interactive online Outlook portal (see https://ukho.ceh.ac.uk/). We will include this justification in the revised manuscript.

Line 499, In some sections, the authors attribute skills in some areas like the south and east to initial hydrological conditions or river memory. Is this based on prior knowledge of basin characteristics?

>> The skill in some areas, such as the south and east, are attributed to strong influence from initial hydrological conditions and catchment memory. As a result, the river flow predictability across these regions for both the standard ESP and HWA methods are high. This region contains mainly groundwater-dominated catchments with high catchment storage. This has been shown in the rigorous skill assessment of the standard ESP method in Harrigan et al., (2018) using the same set of catchments, including relating ESP skill to total catchment storage. Catchments in these regions also yield skilful persistence forecasts (i.e. persistence of flow anomaly from previous month), as shown by Svensson et al. (2016). We will strengthen this statement by referring to past work and knowledge of basin characteristics.

Some other thoughts:

Given HWA's success in winter, would you consider a dynamic framework switch between forecasting methods seasonally (e.g., HWA in winter, other methods in summer)?

>> This is an excellent suggestion. We are currently working on this topic. There is already some discussion of multi-model and multi-method forecast blending in the current discussion section, but we will further highlight the potential of improved performance and reliability of hydrological forecasts by blending forecast products based on forecast skill assessed over a hindcast period. We will also refer readers to Tanguy et al., (2024) which also proposed a similar idea for a "*flexible combinatory system that would dynamically choose the most effective method based on specific factors such as catchment characteristics, time of year and lead time*".

And for summer, are there other alternative indices that might outperform NAO for selecting analogues?

>> Thank you for this suggestion. We agree that a focus on alternative indices could potentially improve summer forecast skill. We have discussed this possibility in the original manuscript's discussion section: *"Improvements in the predictability of the East Atlantic pattern, which exhibits strong influence on rainfall variability, particularly across southern Britain (West et al., 2019), could contribute to further advances in national summer river flow predictability".* We will also make reference to potential improvements in summer flow predictability with better understanding of large-scale atmospheric teleconnections. Recently, Chevuturi et al., (2025) have demonstrated that the predictability of UK summer river flows and drought is linked to variations in North Atlantic sea surface temperatures and its atmospheric teleconnection pathway, which shows signs of predictability at a 1.5 years lag time. This improved understanding of teleconnections can be utilised to improve summer river flow predictability.

Just curious, what is the ratio of autumn/winter rainfall?

>> The ratio of UK average autumn/winter rainfall is 1.04

**References:**

Chevuturi, A., Oltmanns, M., Tanguy, M., Harvey, B., Svensson, C. and Hannaford, J., 2025. Oceanic drivers of UK summer droughts. *Communications Earth & Environment*, *6*(1), p.437.

Harrigan, S., Prudhomme, C., Parry, S., Smith, K. and Tanguy, M., 2018. Benchmarking ensemble streamflow prediction skill in the UK. *Hydrology and Earth System Sciences*, *22*(3), pp.2023-2039.

Svensson, C., 2016. Seasonal river flow forecasts for the United Kingdom using persistence and historical analogues. *Hydrological Sciences Journal*, *61*(1), pp.19-35.

Stringer, N., Knight, J. and Thornton, H., 2020. Improving meteorological seasonal forecasts for hydrological modelling in European winter. *Journal of Applied Meteorology and Climatology*, *59*(2), pp.317-332.

Tanguy, M., Eastman, M., Chevuturi, A., Magee, E., Cooper, E., Johnson, R.H., Facer-Childs, K. and Hannaford, J., 2025. Optimising Ensemble Streamflow Predictions With Bias Correction And Data Assimilation Techniques. *Hydrology and Earth System Sciences*, *29*, pp.1587-1614.

---

## Author Comment (AC2)

**UK Hydrological Outlook using Historic Weather Analogues**

**Reviewer #2**

Thank you for the opportunity to review this manuscript evaluating the use of Historic Weather Analogues (HWAs) for improved seasonal streamflow prediction across UK catchments. This work builds on over 25 years of research on incorporating climate information into seasonal forecasts (e.g., Hamlet and Lettenmaier, 1999), providing a systematic hindcast evaluation and nation-wide case study of the HWA method that is directly relevant to operational prediction (i.e., the UK Hydrologic Outlook). The authors demonstrate how forecasted sea level pressure anomalies from GloSea6 can be used to select HWAs as inputs to the GR6J hydrology model, leading to improved streamflow prediction over traditional ESP methods in regions more influenced by meteorology than initial hydrologic conditions. While the core methodological progress is incremental, the study provides a rigorous benchmarking of the HWA approach against both climatological and ESP baselines, with results showing meaningful wintertime skill improvements.

>> We thank the reviewer for their detailed comments on our manuscript. We are glad the reviewer recognises the importance of our results as a rigorous benchmarking study for the HWA approach against both climatology and ESP approaches for the UK. Please find below our response in red.

**Major Comments**

1. The use of retrospective model simulations (here, termed "simulated observations") in verification rather than actual streamflow observations is unconventional, and not immediately clear. At the very least, this needs to be better described in the methods section, but it should also be disclosed elsewhere. Additionally, I would request that you strongly consider renaming this variable to a more transparent term, e.g. "retrospective simulation", retro-sim), so that it is clear that this is not an observational dataset. Please justify this choice (e.g., incomplete obs. dataset, upstream regulations, etc.) and include a discussion of its limitations in the Discussion section.

>> Studies have adopted different terms for simulated river flows over a baseline observational period, such as "proxy observations" or "retrospective simulation" as the reviewer noted. It is common to assess forecast skill using "simulated observed" river flows rather than a direct comparison against observed river flows (e.g. Pappenberger et al., 2015; Wood et al., 2016; Harrigan et al., 2018). The use of "simulated observed" or "retrospective simulated" river flows to assess forecast skill has the advantage of isolating the forecast skill from hydrological model biases. Our use of "retrospective simulation" also enables comparison with previous hindcast skill assessment of the standard ESP in the UK from Harrigan et al., (2018), which have also used "simulated observed" flows as a comparator to calculate skill scores.

We recognize the potential for misunderstanding and will adopt the reviewer's suggestion to change our terminology to "retrospective simulation" and make sure to mention our use of this as a baseline, rather than actual streamflow observations, throughout the revised manuscript and the discussion section.

2. I encourage the authors to consider greater use of the active voice throughout the Methods section. At times, it was unclear who was performing certain actions, which made it difficult to follow some of your methods. For example, when discussing the hindcasts from the GloSea6 prediction system, it was not always clear whether the subject was the UK Met Office or the authors themselves (e.g., "In addition, retrospective forecasts ('hindcasts') for each meteorological season over the 1993-2016 period are produced, initialised from a subset of dates (1st, 9th and 17th) each month." … and, "Hindcasts were made for each conventional season (DJF - winter, MAM - spring, JJA - summer and SON - autumn)"). Clearer attribution using active voice will make it easier to follow/understand the methodology.

>> Thank you for this suggestion. We will adopt a clearer active voice for the suggested sentences in the methods section.

3. Consider including brief introductory paragraphs at the start of the Results and Discussion sections. Such introductions can outline the key questions addressed, clarify the structure of each section, and provide context for the analyses that follow. It also provides gentler transitions for the reader.

>> Thank you for this suggestion. We will add brief introductory paragraphs to the start of the Results and Discussion sections.

4. Please revise the description of the ESP (and HWA) methods to consistently use standard forecasting terminology such as "meteorological traces," "ensemble members," "hindcast initialization," and "lead time" For example, clarify that ESP ensemble members are generated by running the hydrological model with observed meteorological input sequences (traces) from different years in the historical record, conditioned on the current initial hydrologic state.

>> Thank you for this suggestion. We agree that standard forecasting terminology should be used throughout. We will adopt the terminology suggested by the reviewer throughout the revised manuscript.

It would also be helpful if the authors explicitly describe the GloSea6 climate hindcast initializations (1st, 9th, 17th) to the hydrological forecast initializations (e.g., does each hydrologic forecast correspond to a climate ensemble member initialized on those dates, or are hydrological forecasts always initialized at the start of the season?)

>> Hydrological hindcasts were always initialised from the start of each season. We will clarify that in the revised manuscript.

**Minor Comments**

**Introduction**

Line 39 - "potential risk during flood-prone seasons"

>> Thanks, will change in revised manuscript

Lines 43-47 - This sentence feels incomplete. Please be more explicit about the implications of the dependencies of IHC and seasonal weather predictability on seasonal hydrologic forecasting. Explicitly stating these implications will help set the stage for the rest of the paper.

Line 53 - consider adding commas after each e.g. (e.g., Hulme and Barrow, 1997), here and elsewhere in the manuscript. Also, it may be the case that the e.g. is overused in your citations in Section 1.1

Line 60 - Instead of just the eastern US, this may be more broadly defined as from eastern North America to Scandinavia

Line 64 - consider condense the West et al. (2019) citation to the end of line 66 only, to remove redundancy.

>> Thanks, we will make all the above changes in revised manuscript

Line 68 - Please define SNAO more explicitly

>> We will make clear to define SNAO as summer NAO.

Line 74 - Broaden topic sentence to hydrologic response variability of all catchments across the UK, then hone in to talk about regional difference, e.g., between the SE and NW

>> Thanks, will add the influence of SNAO on different regions of the UK.

Line 88 - Existing approaches for forecasting what precisely? Weather? Streamflow?

>> We will specify streamflow forecasts.

Line 89 -  While the term "analogy" is used, the more common terminology in the literature is "analogue" or "analog" forecasts.

Line 95 - Consider citing the foundational LSTM paper on rainfall-runoff modeling (Kratzert et al 2019)

Line 102 - Please consider citing Day (1985), which documents the original US National Weather Service ESP methodology.

Line 104/105 - break this into two sentences, 1) describing the role of IHCs in providing skill for ESP forecasts (perhaps with more emphasis on this point), and 2) the dominant processes influencing IHCs across the UK

>> Thanks, we will make all the above changes in revised manuscript

Line 129 - Rather than implying that detailed hydrologic modeling is not possible with current NWP output, emphasize the importance of downscaling methods when using NWP as hydrologic model forcing due to discrepancies in spatial resolution

>> Thanks, we will detail the challenges of detailed hydrological modelling using existing generation of NWP outputs.

Section 1.2: This section could benefit from a revised organization. I suggest the following structure:

**1: Simple statistical methods**

**2: ESP-based methods**

**3: Stylised scenario approaches**

**4: NWP-forced hydrologic modelling**

>> Thanks, following similar suggestions from reviewer#1, we will more clearly distinguish "deterministic forecasts" with "ensemble-based approaches", with sub-strands distinguishing statistical methods, ESP-based methods, stylised scenario approaches and NWP-forced modelling.

Additionally, consider the placement of the discussions of LSTMs in this scheme. LSTMs are not traditional statistical methods (as currently categorized, though they are data-driven) and are maybe better thought of as a model type that could be applied within any of the other forecasting approaches. Maybe it would be more appropriate to have a brief discussion of different types of hydrologic models (conceptual, process/physics-oriented, and data-driven including both simple statistical and deep learning methods) all of which can be applied with any of these forecasting methods

>> Thank you for raising this point. We agree that LSTMs should not be considered a separate strand within traditional statistical methods. We will add a brief discussion of the typology of hydrological models and make clear that any one of the hydrological model types could be applied with these forecasting methods.

Line 131 - "climate information into ESP forecasts, often referred to as conditional ESP". Or similar.

>> This will be made clear in the revised structure of Section 1.2

Line 132 - "sub-sampling meteorological traces" (good to define met trace first)

>> Thank you for the suggested text. These will be changed in accordance with standard forecasting terminologies as suggested by the reviewer.

Line 134 - Please discuss the mechanism of improvements found in W&L (2006) and Beckers et al. (2016)

Line 138 - Small typo: "studies have shown"

Line 139 - Define horizon of "long lead times"

Line 155 - missing em dash after (in-prep)

>> Thanks, we will make all the above changes in revised manuscript

Section 1.3: The transition from conditioned ESP approaches to the HWA method is logical but could be made clearer. Consider adding a sentence or two to explicitly link the evolution of methods, e.g.:

*"While conditioned ESP methods rely on sub-sampling or weighting historical traces based on large-scale climate signals, the HWA approach further advances this concept by identifying specific historical weather patterns that closely match forecasted atmospheric circulation states. This enables forecasts to more directly leverage reliable dynamical model outputs and can provide higher spatial resolution than traditional ESP-based methods."*

>> Thanks for the suggested text, we will change this according to the reviewer's suggestion in the revised manuscript.

**Methods**

Line 168 -  I would not capitalize Chalk and Limestone

Line 169/170 - Exactly how many catchments of your study catchments are part of the UK Benchmark Network?

>> Out of the 314 selected catchments, 128 are from the UK Benchmark Network. We will include that in the revised manuscript.

Section 2.1 - Consider breaking this section into two paragraphs, and being more explicit about what was used as a model input vs. simply a descriptive variables/catchment attribute. The current paragraph reads like a dense list of data sources – more context would be helpful.

Line 200 - this should be two sentences.

Line 205 - Citation on the mKGE?

Line 207 - This i.e. seems out of place. Consider removing.

>> Thanks, we will make all the above changes in revised manuscript

Line 207 - GR6J model results from whom/where? Be more explicit please.

>> This relates to the next sentence and Figure 1 showing the GR6J model results across the selected catchments.

Line 229 - This paragraph (and others in this section, perhaps) could benefit from a clearer problem statement as a topic sentence

>> We will add this.

Line 230 - What do you define as high spatial resolution? Or at least, what is the average catchment size? This might help us better understand discrepancies between seasonal climate model outputs and needed hydrology model inputs.

>> We propose to add a table of average catchment size to the supplementary materials. We consider the use of 1km meteorological observations here as high spatial resolution.

Line 238 - not just simulated monthly patterns, but *predicted* monthly patterns from the hindcasts. It is important to make this clear to help the reader with understanding the key method.

>> Thanks – this will be added to make our methodology clear.

Line 245 - Similarly, this paragraph would benefit from first defining the problem statement, e.g., that the signal to noise of NAO in seasonal climate systems is too small during the winter, and then discussing how you address this challenge.

>> We will add an introductory sentence to this paragraph defining the signal to noise problem as suggested

Section 2.4.1 - Is this a daily timestepped model? Line 274 suggests so, but please state.

>> We will make clear we have used a daily hydrological model.

Line 272 - It would be helpful to use more conventional terms to describe your ESP approach, as you did in Section 1.2. For example, you could write: "For each month in the hindcast period, three-month lead time seasonal ESP hindcasts were generated using the GR6J model forced with meteorological traces from the historical observation record." This would make your methods more transparent and easier to follow for readers familiar with ESP-based forecasting.

>> This will be changed following the reviewer's encouragement to adopt standard forecast terminology.

Section 2.4.2 - Consider breaking into two paragraphs for improved readability

Section 2.5 - Consider breaking into two or more paragraphs for improved readability

>> Both suggestions will be made.

**Results**

Line 229 - At the beginning of a section, it may be helpful to be specific about what types of forecasts (and at what lead times) you are talking about, e.g. "seasonal streamflow forecasts", not weather forecasts (for example)

Line 330 - Have you defined this positive skill threshold of 0.05 yet?

Line 346 - NI? Not defined.

Line 350 - Use proper name and then define abbreviation

>> We will make all the above changes in revised manuscript

Figure 3 - Consider including key details to allow figures to stand alone more effectively, e.g., that this is across 314 UK study catchments. Same for hindcast period years. Also, add a CRPSS label to the colorbars, and a key for the arrow direction.

>> We will add that to the figure legend. The colors and arrow direction are already included in the figure caption.

Figure 4 - Add colorbar labels. In the caption, consider revising to: "Blue colours indicate the HWA method is better than the ESP benchmark reference, red colours…" Additionally, please clarify whether the symbol direction (triangle up/down) represents the same information as the color (i.e., skill difference). If not, consider using the symbol to convey complementary information – such as the sign of the HWA skill score relative to climatology. For example, an upward triangle could indicate HWA is skillful compared to climatology, while a downward triangle could indicate it is not. This would allow readers to quickly assess not only where HWA outperforms ESP, but also where it is meaningfully skillful in an absolute sense.

Figure 5 - Add a descriptive label on colorbar

Figure 6 - Please consider updating the y-axis label to "DJF Mean Daily Flow" or a similar more descriptive term. Additionally, consider adding a shaded region indicating the 25th–75th percentile range around the ESP and HWA ensemble means. This would provide a sense of ensemble dispersion and improve the interpretability of the forecast spread. Please also update "Obs Sim" to "Retro Sim".

>> All of the above will be changed to improve figure clarity as the reviewer suggested.

Figure 7 - Is the colorbar incorrectly labeled (e.g. AUC instead of ROC)?

>> Yes, this was mislabelled – we will correct.

Line 425 - Develop this logic just a bit further – why are you highlighting winter flow predictability over other seasons?

>> We have chosen to highlight winter river flow predictability as it is the season where the highest number of catchments saw improved flow predictability against the standard ESP method. We also highlight the winter season as rainfall over the UK is strongly positively correlated with the leading mode of climate variability (i.e. winter NAO). In other seasons, total rainfall variability is less well explained by the leading modes of climate variability (e.g. summer NAO in JJA). The influence of global climate patterns on UK weather in the other seasons also tends to be smaller than in the winter. Hence, we would expect a forecasting system conditioned on predictability of

weather patterns to have a higher increase in skill in the winter compared to other seasons.

Figure 9 - The probability bar plots are hard to see and even harder to compare against the observed outlook categories. That said, I do think the case studies are really valuable, so I think it's worth considering how to improve this figure. What about colored pie charts? Or aggregation of results to regions? I would also define the NAO phase in the "Winter 1994/95" and "Winter 2009/10" subtitles.

>> We would prefer to retain this figure as is. This is the standard format for visualising forecast results from the UK Hydrological Outlook. The visualisation approach and the colours have been adopted after extensive stakeholder consultation and is operationalised via both the monthly Hydrological Outlook and in the interactive online Outlook portal. We will define the NAO phase in the revised manuscript.

Figure 10 - Please clarify the scientific value of including jet speed as an intermediate variable in this figure. Does examining jet speed, in addition to NAO index, provide insight into the added value of HWA versus ESP?

>> We have included jet speed as it is a more direct description of atmospheric circulation compared to the NAO. As the HWA approach do not sample for analogues using the NAO index, but instead selects analogues based on the spatial MSLP pattern, a more process-based variable like jet stream indicators, adds more physical insights to the HWA approach. Jet speed is a well-known driver of UK rainfall and is highly correlated with winter rainfall, particularly over western Scotland (with a stronger correlation than winter NAO).

Additionally, is there a statistically significant difference in HWA-ESP skill between different NAO phases? Might you be able to explain differences in skill between the two methods using total catchment storage (e.g., Harrigan et al. (2018), line 503)?

>> As stated in our results section, we think there is not enough years within the hindcast ensemble (1993-2016) to robustly determine whether there are statistically significant difference in HWA-ESP skill between different winter NAO phases. It has been shown robustly by Harrigan et al., (2018) that ESP skill is associated with total catchment storage. The HWA approach, as a conditioned ESP approach, would naturally retain skill in areas with high catchment storage (e.g. as shown by year-round skill in summer flow predictability for catchments in the south-east), but also leverage the improved predictability of atmospheric circulation patterns to improve skill in catchments where river flow variability is strongly tied with rainfall variability (e.g. those with limited catchment storage). We will strengthen this argument in the revised discussion section.

**Discussion**

Line 502 - Please clarify that these conclusions about the role of IHCs are specific to the UK context, as the cited studies (Svensson, 2016; Svensson et al., 2015) are focused on UK catchments. Otherwise, broaden citations.

>> We will specify these conclusions are for the UK context.

Line 510 - What was the study domain of the Baker et al. (2018) study?

>> This was a global study investigating the physical drivers contributing to the predictability of the winter NAO in various forecasting systems (including GloSea).

Section 4.2 - The current text details the comparative skill of HWA and ESP methods, but could be strengthened by more explicitly discussing the implications of the role of IHCs in ESP forecasts. Please elaborate on what your findings suggest about when and where IHCs are most critical for skill, how this influences forecast design and operational use, and what this means for improving seasonal prediction in regions dominated by IHC versus meteorological predictability.

>> Thank you for the suggestion. We will discuss clearly when and where IHCs contribute the most to forecast skill and when/where improved meteorological predictability contributes to forecast skill. This is also in accordance with similar suggestions from reviewer #1 with more discussion on forecast design, such as multi-method forecast blending based on forecast skill assessed over a hindcast period (e.g. choosing the best forecasting method for different seasons).

**Conclusion**

Line 629 - Suggested addition: "... in South East England, where initial hydrologic conditions related to groundwater storage provide seasonal predictability".

>> We will add this.

**References:**

Harrigan, S., Prudhomme, C., Parry, S., Smith, K. and Tanguy, M., 2018. Benchmarking ensemble streamflow prediction skill in the UK. *Hydrology and Earth System Sciences*, 22(3), pp.2023-2039.

Pappenberger, F., Ramos, M.H., Cloke, H.L., Wetterhall, F., Alfieri, L., Bogner, K., Mueller, A. and Salamon, P., 2015. How do I know if my forecasts are better? Using benchmarks in hydrological ensemble prediction. *Journal of Hydrology*, 522, pp.697-713.

Svensson, C., 2016. Seasonal river flow forecasts for the United Kingdom using persistence and historical analogues. *Hydrological Sciences Journal*, 61(1), pp.19-35.

Stringer, N., Knight, J. and Thornton, H., 2020. Improving meteorological seasonal forecasts for hydrological modelling in European winter. *Journal of Applied Meteorology and Climatology*, *59*(2), pp.317-332.

Wood, A.W., Hopson, T., Newman, A., Brekke, L., Arnold, J. and Clark, M., 2016. Quantifying streamflow forecast skill elasticity to initial condition and climate prediction skill. *Journal of Hydrometeorology*, *17*(2), pp.651-668.

---

## Author Response (AR1)

**UK Hydrological Outlook using Historic Weather Analogues**

**Response to editor**

I would like to draw your attention to one specific point that aligns with the reviewers' suggestions- namely, the incorporation of streamflow observations in addition to the model-based simulations. While I understand your response of using retrospective model simulations from the model benchmarking point of view, I still encourage you to also include a verification assessment using observed streamflow data. Given the availability of a substantial number of gauging stations with long-term records across UK catchments, integrating such an observational-based analysis will allow for a more robust assessment of the actual skill of the proposed Historic Weather Analogues (HWAs) compared to other approaches. I therefore suggest including this analysis which can further strengthen the overall message of this work.

>> We thank the editor for handling our manuscript. We have now included a verification assessment against observed river flow data in the revised manuscript with an additional figure in the supplementary materials (Figure S2 and response to reviewer 2 below) and associated text. Please find below our response to the reviewers with relevant line numbers in the tracked changes revised manuscript.

In the process of revision, we have uncovered an error in the original standard ESP results where the leave-three-out-cross-validation procedure we intended to apply was not applied correctly. As a result, the standard ESP hindcast skill scores were incorrectly inflated (i.e. the hindcast year itself was not removed within the probabilistic ensemble). We have corrected this in the new version and the skill scores for the standard ESP are now a fairer reflection of forecast skill. The main results and narrative of the paper remain unchanged. We apologise for this error.

**Response to reviewers**

**Reviewer #1**

Thank you for this well-written and timely manuscript, which describes a new experiment in the UK national hydrological forecasting system. The method incorporates historical observations from analogue months by matching the large-scale circulation patterns, and use them as forcings to generate hydrological forecasts. The study shows improvements in seasonal forecasts skills and event categorization, particularly in winter (the rainy season). This work is both important and valuable for the hydrological forecast community.

>> We thank the reviewer for their detailed comments on our manuscript. We are glad the reviewer recognises the importance of our results and the value of the methodology for the hydrological forecast community. Please find below our response in red.

Below are some comments to further discuss the idea with the authors and improve readability:

Line 68, Consider specifying "summer NAO (SNAO)" when first time refer to it.

>> Thanks – this has been changed.

Line 86, section 1.2, The section mentions four forecast categories, but the introduction states there are "three strands." Please clarify this. And the methods in the first category might be better to conclude as "descriptive forecasts" to distinguish them from ensemble-based approaches that come later.

>> Thank you for noticing our error. We have noted difference between deterministic and ensemble-based forecasts in our revised manuscript. We have also aligned this section with standard forecasting terminology.

Line 144, Are analogue months considered independently (i.e., monthly NAO indices)? Have you tested using moving-window averages for NAO to account for variability in selecting analogues?

>> Analogue months are considered independently. In the winter seasonal forecasts (i.e. DJF), the monthly NAO values are accounted for in the analogue selection procedure. However, when matching MSLP patterns, only the seasonal mean MSLP pattern were used for analogue selection based on spatial similarity of the forecast MSLP pattern. We have clarified this in the revised manuscript.

Line 220, Could you clarify why 17 ensemble members were chosen here? A flow chart illustrating the selection process would be helpful.

>> There are currently 17 hindcast members available for each of the three initialisation dates (i.e. giving 51 ensemble members per season). The resulting 51 ensemble members is broadly similar in size to the operational forecast so is considered a fair reflection of the operational forecast skill. We have pointed readers to Stringer et al., (2020) which provides the details of the hindcast ensemble and analogue selection methodology and have revised this section to reflect similarity with the operational system (Lines 278-280).

Line 235, For analogue season selection, have you plotted rainfall patterns for an example season to assess consistency among analogue months? It would be interesting to see such visualization (e.g., a map or time series).

>> We have included the below map showing correlation between HWA ensemble mean rainfall and observed mean rainfall for the different seasons in the supplementary materials of the revised manuscript.

We further note that our companion paper (Rhodes-Smith et al., in review - https://doi.org/10.5194/egusphere-2025-2506) includes a similar figure showing correlation of hindcast ensemble mean rainfall and observed rainfall across Great Britain, but also includes ungauged locations (figure 2 in their paper).

[Figure]

**Figure 1** Correlation between HWA forecast ensemble mean and HadUK-Grid observed seasonal rainfall across 314 UK catchments over the hindcast period (1993-2016).

Line 336, The text here continues analyzing results from Figure 3. But it reads like it is from Figure S1. Just specify it would help.

>> We have specified Figure 3.

Line 341, What does "heterogeneity" refer to here, between the areas or between the methods? How are the numbers reflecting heterogeneity, could you explain a bit more.

>> Heterogeneity here just means large variation in CRPSS values within the different hydroclimate regions. We will rephrase to "there is substantial variation in CRPSS values for catchments within each hydroclimate region"

Line 348, consider adding the catchment numbers together with the ratio, e.g. XX out of YY.

>> Thank you for the suggestion. We have added the catchment numbers along with the ratio (L441-442).

Line 388, Typo: "--0.38" should likely be "-0.38." Is this value statistically significant?

>> Thank you for spotting the typo, we have corrected.

Line 420, Figure 7, This is an excellent visualization. I also noticed that for summer, both high flow events and low flow events had a drop in performance using HWA. Could this reflect challenges in low-flow forecasting? Since later in the discussion the authors mentioned summer is a future target, so maybe already mention it here while discussing the results for summer months.

>> Thank you for your comments on the figure visualisation. The reviewer is correct that summer flow predictability for summer months have dropped when using the HWA method, mainly outside of groundwater-dominated, slow-responding catchments in the south and east. We agree that this is an on-going challenge for low flow forecasting

and have mentioned this particular challenge in the revised manuscript results section (L521-523).

Line 452, Is it better to show the correct ratio for each station instead of the full distribution? Or if distributions are preferred, please just specify the reasons.

>> We have adopted to show the full distribution as this is the standard format for visualising forecast results from the UK Hydrological Outlook. The visualisation approach and the colours have been adopted after extensive stakeholder consultation and is operationalised via both the monthly Hydrological Outlook and in the interactive online Outlook portal (see https://ukho.ceh.ac.uk/). We have included this justification in the revised manuscript as part of the revised figure caption.

Line 499, In some sections, the authors attribute skills in some areas like the south and east to initial hydrological conditions or river memory. Is this based on prior knowledge of basin characteristics?

>> The skill in some areas, such as the south and east, are attributed to strong influence from initial hydrological conditions and catchment memory. As a result, the river flow predictability across these regions for both the standard ESP and HWA methods are high. This region contains mainly groundwater-dominated catchments with high catchment storage. This has been shown in the rigorous skill assessment of the standard ESP method in Harrigan et al., (2018) using the same set of catchments, including relating ESP skill to total catchment storage. Catchments in these regions also yield skilful persistence forecasts (i.e. persistence of flow anomaly from previous month), as shown by Svensson et al. (2016). We have strengthened this statement by referring to past work and knowledge of basin characteristics (Section 4.1).

Some other thoughts:

Given HWA's success in winter, would you consider a dynamic framework switch between forecasting methods seasonally (e.g., HWA in winter, other methods in summer)?

>> This is an excellent suggestion. We are currently working on this topic. There is already some discussion of multi-model and multi-method forecast blending in the current discussion section, but have further highlighted the potential of improved performance and reliability of hydrological forecasts by blending forecast products based on forecast skill assessed over a hindcast period. We also referred readers to Tanguy et al., (2024) which also proposed a similar idea for a "*flexible combinatory system that would dynamically choose the most effective method based on specific factors such as catchment characteristics, time of year and lead time*" (L755-759).

And for summer, are there other alternative indices that might outperform NAO for selecting analogues?

>> Thank you for this suggestion. We agree that a focus on alternative indices could potentially improve summer forecast skill. We have discussed this possibility in the

original manuscript's discussion section: *"Improvements in the predictability of the East Atlantic pattern, which exhibits strong influence on rainfall variability, particularly across southern Britain (West et al., 2019), could contribute to further advances in national summer river flow predictability".* We now also make reference to potential improvements in summer flow predictability with better understanding of large-scale atmospheric teleconnections. Recently, Chevuturi et al., (2025) have demonstrated that the predictability of UK summer river flows and drought is linked to variations in North Atlantic sea surface temperatures and its atmospheric teleconnection pathway, which shows signs of predictability at a 1.5 years lag time. This improved understanding of teleconnections can be utilised to improve summer river flow predictability (736-739).

Just curious, what is the ratio of autumn/winter rainfall?

>> The ratio of UK average autumn/winter rainfall is 1.04

**Reviewer #2**

Thank you for the opportunity to review this manuscript evaluating the use of Historic Weather Analogues (HWAs) for improved seasonal streamflow prediction across UK catchments. This work builds on over 25 years of research on incorporating climate information into seasonal forecasts (e.g., Hamlet and Lettenmaier, 1999), providing a systematic hindcast evaluation and nation-wide case study of the HWA method that is directly relevant to operational prediction (i.e., the UK Hydrologic Outlook). The authors demonstrate how forecasted sea level pressure anomalies from GloSea6 can be used to select HWAs as inputs to the GR6J hydrology model, leading to improved streamflow prediction over traditional ESP methods in regions more influenced by meteorology than initial hydrologic conditions. While the core methodological progress is incremental, the study provides a rigorous benchmarking of the HWA approach against both climatological and ESP baselines, with results showing meaningful wintertime skill improvements.

>> We thank the reviewer for their detailed comments on our manuscript. We are glad the reviewer recognises the importance of our results as a rigorous benchmarking study for the HWA approach against both climatology and ESP approaches for the UK. Please find below our response in red.

**Major Comments**

1. The use of retrospective model simulations (here, termed "simulated observations") in verification rather than actual streamflow observations is unconventional, and not immediately clear. At the very least, this needs to be better described in the methods section, but it should also be disclosed elsewhere. Additionally, I would request that you strongly consider renaming this variable to a more transparent term, e.g. "retrospective simulation", retrosim), so that it is clear that this is not an observational dataset. Please justify this choice (e.g., incomplete obs. dataset, upstream regulations, etc.) and include a discussion of its limitations in the Discussion section.

>> Studies have adopted different terms for simulated river flows over a baseline observational period, such as "proxy observations" or "retrospective simulation" as the reviewer noted. It is common to assess forecast skill using "simulated observed" river flows rather than a direct comparison against observed river flows (e.g. Pappenberger et al., 2015; Wood et al., 2016; Harrigan et al., 2018). The use of "simulated observed" or "retrospective simulated" river flows to assess forecast skill has the advantage of isolating the forecast skill from hydrological model biases. Our use of "retrospective simulation" also enables comparison with previous hindcast skill assessment of the standard ESP in the UK from Harrigan et al., (2018), which have also used "simulated observed" flows as a comparator to calculate skill scores, as now noted in the revised manuscript (L378-382). We recognize the potential for misunderstanding and have adopt the reviewer's suggestion to change our terminology to "retrospective simulation" and have denoted that we used this this as a baseline, rather than actual streamflow observations, throughout the revised manuscript.

To further address the reviewer's concern, we have added a new figure in the supplementary materials of CRPSS calculated relative to actual observed river flows for all seasons (see new supplementary Figure S2 in revised manuscript and below) and have referred to it in the revised manuscript (L433-436). We note that the broad spatial pattern of hindcast skill remains very similar to Figure 3 although at individual catchments, the raw CRPSS values can be different given possible systematic bias in the hydrological model simulations and, potentially cases where certain years within the hindcast period have missing observational data.

[Figure]

**Figure 2** Probabilistic hindcast skill for the ESP (top) and HWA (bottom) methods across the hindcast period (1993-2016) for 314 UK catchments. The metric used is the CRPSS, and is calculated for the hindcast period by comparing HWA and ESP with benchmark climatology (observed daily river flows) per season. Blue colours indicate the historic weather analogues method has higher skill than the benchmark climatology (red colours show the historic weather analogues method is worse than climatology). White colours indicate neutrally skilful forecasts. The direction of the symbol indicates the sign of the respective skill score.

2. I encourage the authors to consider greater use of the active voice throughout the Methods section. At times, it was unclear who was performing certain actions, which made it difficult to follow some of your methods. For example, when discussing the hindcasts from the GloSea6 prediction system, it was not always clear whether the subject was the UK Met Office or the authors themselves (e.g., "In addition, retrospective forecasts ('hindcasts') for each meteorological season over the 1993-2016 period are produced, initialised from a subset of dates (1st, 9th and 17th) each month." … and, "Hindcasts were made for each conventional season (DJF - winter, MAM - spring, JJA - summer and SON - autumn)"). Clearer attribution using active voice will make it easier to follow/understand the methodology.

>> Thank you for this suggestion. We will adopt a clearer active voice for the suggested sentences in the methods section.

3. Consider including brief introductory paragraphs at the start of the Results and Discussion sections. Such introductions can outline the key questions

addressed, clarify the structure of each section, and provide context for the analyses that follow. It also provides gentler transitions for the reader.

>> Thank you for this suggestion. We have added introductory paragraphs to the start of the Results (L411-415) and Discussion (L612-615) sections.

4. Please revise the description of the ESP (and HWA) methods to consistently use standard forecasting terminology such as "meteorological traces," "ensemble members," "hindcast initialization," and "lead time" For example, clarify that ESP ensemble members are generated by running the hydrological model with observed meteorological input sequences (traces) from different years in the historical record, conditioned on the current initial hydrologic state.

>> Thank you for this suggestion. We agree that standard forecasting terminology should be used throughout. We have adopted standard terminology as suggested by the reviewer throughout the revised manuscript.

It would also be helpful if the authors explicitly describe the GloSea6 climate hindcast initializations (1st, 9th, 17th) to the hydrological forecast initializations (e.g., does each hydrologic forecast correspond to a climate ensemble member initialized on those dates, or are hydrological forecasts always initialized at the start of the season?)

>> Hydrological hindcasts were always initialised from the start of each season. We will clarify that in the revised manuscript.

**Minor Comments**

**Introduction**

Line 39 - "potential risk during flood-prone seasons"

>> Corrected.

Lines 43-47 - This sentence feels incomplete. Please be more explicit about the implications of the dependencies of IHC and seasonal weather predictability on seasonal hydrologic forecasting. Explicitly stating these implications will help set the stage for the rest of the paper.

Line 53 - consider adding commas after each e.g. (e.g., Hulme and Barrow, 1997), here and elsewhere in the manuscript. Also, it may be the case that the e.g. is overused in your citations in Section 1.1

Line 60 - Instead of just the eastern US, this may be more broadly defined as from eastern North America to Scandinavia

Line 64 - consider condense the West et al. (2019) citation to the end of line 66 only, to remove redundancy.

>> Thanks, we have made all the above changes.

Line 68 - Please define SNAO more explicitly

>> We have defined the SNAO acronym in the first appearance.

Line 74 - Broaden topic sentence to hydrologic response variability of all catchments across the UK, then hone in to talk about regional difference, e.g., between the SE and NW

>> We have broadened the topic sentence for this paragraph.

Line 88 - Existing approaches for forecasting what precisely? Weather? Streamflow?

>> We have specified streamflow forecasting.

Line 89 - While the term "analogy" is used, the more common terminology in the literature is "analogue" or "analog" forecasts.

>> We have amended to "analogue"

Line 95 - Consider citing the foundational LSTM paper on rainfall-runoff modeling (Kratzert et al 2019)

>> Now cited.

Line 102 - Please consider citing Day (1985), which documents the original US National Weather Service ESP methodology.

>> Now cited.

Line 104/105 - break this into two sentences, 1) describing the role of IHCs in providing skill for ESP forecasts (perhaps with more emphasis on this point), and 2) the dominant processes influencing IHCs across the UK

>> We have amended this sentence as suggested.

Line 129 - Rather than implying that detailed hydrologic modeling is not possible with current NWP output, emphasize the importance of downscaling methods when using NWP as hydrologic model forcing due to discrepancies in spatial resolution

>> Thank you for the suggestion. We have now made it clearer that downscaling methods are required when using raw NWP outputs in hydrological modelling.

Section 1.2: This section could benefit from a revised organization. I suggest the following structure:

**1: Simple statistical methods**

**2: ESP-based methods**

**3: Stylised scenario approaches**

**4: NWP-forced hydrologic modelling**

>> We have adopted the suggested structure for section 1.2

Additionally, consider the placement of the discussions of LSTMs in this scheme. LSTMs are not traditional statistical methods (as currently categorized, though they are data-driven) and are maybe better thought of as a model type that could be applied within any of the other forecasting approaches. Maybe it would be more appropriate to have a brief discussion of different types of hydrologic models (conceptual, process/physics-oriented, and data-driven including both simple statistical and deep learning methods) all of which can be applied with any of these forecasting methods

>> Thank you for raising this point. We agree that LSTMs should not be considered a separate strand within traditional statistical methods. We have added a brief discussion of the typology of hydrological models and made clear that any one of the hydrological model types could be applied with these forecasting methods (Section 1.2 in revised manuscript).

Line 131 - "climate information into ESP forecasts, often referred to as conditional ESP". Or similar.

>> This was added as suggested.

Line 132 - "sub-sampling meteorological traces" (good to define met trace first)

>> Thank you for the suggested text. These was changed in accordance with standard forecasting terminologies as suggested by the reviewer.

Line 134 - Please discuss the mechanism of improvements found in W&L (2006) and Beckers et al. (2016)

>> Both studies show that forecast skill were improved after sub-sampling for ESP meteorological traces conditional on the current ENSO phase, particularly at catchments that are most directly affected by different large-scale modes of climate variability (like ENSO). We have made this clearer in the revised manuscript.

Line 138 - Small typo: "studies have shown"

Line 139 - Define horizon of "long lead times"

Line 155 - missing em dash after (in-prep)

>> Thanks, we have made all the above changes.

Section 1.3: The transition from conditioned ESP approaches to the HWA method is logical but could be made clearer. Consider adding a sentence or two to explicitly link the evolution of methods, e.g.:

*"While conditioned ESP methods rely on sub-sampling or weighting historical traces based on large-scale climate signals, the HWA approach further advances this concept by identifying specific historical weather patterns that closely match forecasted atmospheric circulation states. This enables forecasts to more directly*

*leverage reliable dynamical model outputs and can provide higher spatial resolution than traditional ESP-based methods."*

>> Thanks for the suggested text, we have added the suggested text in the revised manuscript.

**Methods**

Line 168 - I would not capitalize Chalk and Limestone

>> This is in accordance with normal practice when communicating the UK Hydrological Outlook and Hydrological Summaries. We will prefer to retain the capitalisation.

Line 169/170 - Exactly how many catchments of your study catchments are part of the UK Benchmark Network?

>> Out of the 314 selected catchments, 128 are from the UK Benchmark Network. We have now included this in the revised manuscript.

Section 2.1 - Consider breaking this section into two paragraphs, and being more explicit about what was used as a model input vs. simply a descriptive variables/catchment attribute. The current paragraph reads like a dense list of data sources – more context would be helpful.

>> We have updated this paragraph and breaking it into model input vs descriptive catchment attributes as the reviewer suggested.

Line 200 - this should be two sentences.

Line 205 - Citation on the mKGE?

Line 207 - This i.e. seems out of place. Consider removing.

>> Thanks, we have made all the above changes in revised manuscript

Line 207 - GR6J model results from whom/where? Be more explicit please.

>> This relates to the next sentence and Figure 1 showing the GR6J model results across the selected catchments. We have amended to make this clearer.

Line 229 - This paragraph (and others in this section, perhaps) could benefit from a clearer problem statement as a topic sentence

>> We have added a topic sentence to this section.

Line 230 - What do you define as high spatial resolution? Or at least, what is the average catchment size? This might help us better understand discrepancies between seasonal climate model outputs and needed hydrology model inputs.

>> We propose to reproduce table 1 from Harrigan et al. (2018) which shows several summary statistics (including catchment area) of the same set of catchments. We

consider the use of 1km meteorological observations here as high spatial resolution and have specified this in the revised manuscript.

Line 238 - not just simulated monthly patterns, but *predicted* monthly patterns from the hindcasts. It is important to make this clear to help the reader with understanding the key method.

>> Thanks – this was added to make our methodology clear.

Line 245 - Similarly, this paragraph would benefit from first defining the problem statement, e.g., that the signal to noise of NAO in seasonal climate systems is too small during the winter, and then discussing how you address this challenge.

>> We have added an introductory sentence to this paragraph defining the signal to noise problem as suggested.

Section 2.4.1 - Is this a daily timestepped model? Line 274 suggests so, but please state.

>> We have used a daily hydrological model.

Line 272 - It would be helpful to use more conventional terms to describe your ESP approach, as you did in Section 1.2. For example, you could write: "For each month in the hindcast period, three-month lead time seasonal ESP hindcasts were generated using the GR6J model forced with meteorological traces from the historical observation record." This would make your methods more transparent and easier to follow for readers familiar with ESP-based forecasting.

>> This has been adapted similar to what the reviewer suggested (L335-337).

Section 2.4.2 - Consider breaking into two paragraphs for improved readability

Section 2.5 - Consider breaking into two or more paragraphs for improved readability

>> Both suggestions have been made.

**Results**

Line 229 - At the beginning of a section, it may be helpful to be specific about what types of forecasts (and at what lead times) you are talking about, e.g. "seasonal streamflow forecasts", not weather forecasts (for example)

Line 330 - Have you defined this positive skill threshold of 0.05 yet?

Line 346 - NI? Not defined.

Line 350 - Use proper name and then define abbreviation

>> We have made all the above changes in revised manuscript

Figure 3 - Consider including key details to allow figures to stand alone more effectively, e.g., that this is across 314 UK study catchments. Same for hindcast period years. Also, add a CRPSS label to the colorbars, and a key for the arrow direction.

>> We have amended all figure captions

Figure 4 - Add colorbar labels. In the caption, consider revising to: "Blue colours indicate the HWA method is better than the ESP benchmark reference, red colours…" Additionally, please clarify whether the symbol direction (triangle up/down) represents the same information as the color (i.e., skill difference). If not, consider using the symbol to convey complementary information – such as the sign of the HWA skill score relative to climatology. For example, an upward triangle could indicate HWA is skillful compared to climatology, while a downward triangle could indicate it is not. This would allow readers to quickly assess not only where HWA outperforms ESP, but also where it is meaningfully skillful in an absolute sense.

Figure 5 - Add a descriptive label on colorbar

>> The above suggestions have been changed to improve figure clarity as the reviewer suggested. The colors and arrow direction are already included in the original figure caption.

Figure 6 - Please consider updating the y-axis label to "DJF Mean Daily Flow" or a similar more descriptive term. Additionally, consider adding a shaded region indicating the 25th–75th percentile range around the ESP and HWA ensemble means. This would provide a sense of ensemble dispersion and improve the interpretability of the forecast spread. Please also update "Obs Sim" to "Retro Sim".

>> We have updated the axis labels and added the $25^{th}$-$75^{th}$ percentiles range of the HWA spread. "Obs Sim" is amended to "Retro Sim" in all cases.

Figure 7 - Is the colorbar incorrectly labeled (e.g. AUC instead of ROC)?

>> Yes, this was mislabelled – we have corrected this

Line 425 - Develop this logic just a bit further – why are you highlighting winter flow predictability over other seasons?

>> We have chosen to highlight winter river flow predictability as it is the season where the highest number of catchments saw improved flow predictability against the standard ESP method. We also highlight the winter season as rainfall over the UK is strongly positively correlated with the leading mode of climate variability (i.e. winter NAO). In other seasons, total rainfall variability is less well explained by the leading modes of climate variability (e.g. summer NAO in JJA). The influence of global climate patterns on UK weather in the other seasons also tends to be smaller than in the winter. Hence, we would expect a forecasting system conditioned on predictability of weather patterns to have a higher increase in skill in the winter compared to other seasons. We have expanded on this in Section 4.2 in the revised manuscript.

Figure 9 - The probability bar plots are hard to see and even harder to compare against the observed outlook categories. That said, I do think the case studies are really valuable, so I think it's worth considering how to improve this figure. What about colored pie charts? Or aggregation of results to regions? I would also define the NAO phase in the "Winter 1994/95" and "Winter 2009/10" subtitles.

>> We would prefer to retain this figure as is. This is the standard format for visualising forecast results from the operational UK Hydrological Outlook, the interactive online portal (ukho.ceh.ac.uk) and the routine monthly UK Hydrological Summaries. The visualisation approach and the colours have been adopted after extensive stakeholder consultation and is operationalised via both the monthly Hydrological Outlook and in the interactive online Outlook portal. We have highlighted this in the revised figure caption.

We have added the NAO phase on Figure 9 in the revised manuscript.

Figure 10 - Please clarify the scientific value of including jet speed as an intermediate variable in this figure. Does examining jet speed, in addition to NAO index, provide insight into the added value of HWA versus ESP?

>> We have included jet speed as it is a more direct description of atmospheric circulation compared to the NAO. As the HWA approach do not sample for analogues using the NAO index, but instead selects analogues based on the spatial MSLP pattern, a more process-based variable like jet stream indicators, adds more physical insights to the HWA approach. Jet speed is a well-known driver of UK rainfall and is highly correlated with winter rainfall, particularly over western Scotland (with a stronger correlation than winter NAO). We have added some additional justification in the description of Figure 10 (L591-594).

Additionally, is there a statistically significant difference in HWA-ESP skill between different NAO phases?

>> As stated in our results section, we think there is not enough years within the hindcast ensemble (1993-2016) to robustly determine whether there is statistically significant difference in HWA-ESP skill between different winter NAO phases.

Might you be able to explain differences in skill between the two methods using total catchment storage (e.g., Harrigan et al. (2018), line 503)?

>> It has been shown robustly by Harrigan et al., (2018) that ESP skill is associated with total catchment storage. The HWA approach, as a conditioned ESP approach, would naturally retain skill in areas with high catchment storage (e.g. as shown by year-round skill in summer flow predictability for catchments in the south-east), but also leverage the improved predictability of atmospheric circulation patterns to improve skill in catchments where river flow variability is strongly tied with rainfall variability (e.g. those with limited catchment storage). We have expanded on this in the discussion (L655-662).

**Discussion**

Line 502 - Please clarify that these conclusions about the role of IHCs are specific to the UK context, as the cited studies (Svensson, 2016; Svensson et al., 2015) are focused on UK catchments. Otherwise, broaden citations.

>> We have specified these conclusions for the UK context.

Line 510 - What was the study domain of the Baker et al. (2018) study?

>> This was a study investigating the global physical drivers contributing to the predictability of the winter NAO across the European domain in various forecasting systems (including GloSea). We have specified that in the revised manuscript.

Section 4.2 - The current text details the comparative skill of HWA and ESP methods, but could be strengthened by more explicitly discussing the implications of the role of IHCs in ESP forecasts. Please elaborate on what your findings suggest about when and where IHCs are most critical for skill, how this influences forecast design and operational use, and what this means for improving seasonal prediction in regions dominated by IHC versus meteorological predictability.

>> Thank you for the suggestion. We have now discussed when and where IHCs contribute the most to forecast skill and when/where improved meteorological predictability contributes to forecast skill. This is in addition to expanded discussion in response to reviewer #1 on forecast design, such as multi-method forecast blending based on forecast skill assessed over a hindcast period (e.g. choosing the best forecasting method for different seasons) (L753-756).

**Conclusion**

Line 629 - Suggested addition: "... in South East England, where initial hydrologic conditions related to groundwater storage provide seasonal predictability".

>> We have added this.

**References:**

Chevuturi, A., Oltmanns, M., Tanguy, M., Harvey, B., Svensson, C. and Hannaford, J., 2025. Oceanic drivers of UK summer droughts. *Communications Earth & Environment*, *6*(1), p.437.

Harrigan, S., Prudhomme, C., Parry, S., Smith, K. and Tanguy, M., 2018. Benchmarking ensemble streamflow prediction skill in the UK. *Hydrology and Earth System Sciences*, *22*(3), pp.2023-2039.

Pappenberger, F., Ramos, M.H., Cloke, H.L., Wetterhall, F., Alfieri, L., Bogner, K., Mueller, A. and Salamon, P., 2015. How do I know if my forecasts are better? Using

benchmarks in hydrological ensemble prediction. *Journal of Hydrology*, *522*, pp.697-713.

Svensson, C., 2016. Seasonal river flow forecasts for the United Kingdom using persistence and historical analogues. *Hydrological Sciences Journal*, *61*(1), pp.19-35.

Stringer, N., Knight, J. and Thornton, H., 2020. Improving meteorological seasonal forecasts for hydrological modelling in European winter. *Journal of Applied Meteorology and Climatology*, *59*(2), pp.317-332.

Tanguy, M., Eastman, M., Chevuturi, A., Magee, E., Cooper, E., Johnson, R.H., Facer-Childs, K. and Hannaford, J., 2025. Optimising Ensemble Streamflow Predictions With Bias Correction And Data Assimilation Techniques. *Hydrology and Earth System Sciences*, *29*, pp.1587-1614.

Wood, A.W., Hopson, T., Newman, A., Brekke, L., Arnold, J. and Clark, M., 2016. Quantifying streamflow forecast skill elasticity to initial condition and climate prediction skill. *Journal of Hydrometeorology*, *17*(2), pp.651-668.